# FINE-GRAINED ITERATIVE ADVERSARIAL ATTACKS WITH LIMITED COMPUTATION BUDGET

**Zhichao Hou**[1], **Weizhi Gao**[1], **Xiaorui Liu**[1]
[1]North Carolina State University
`{zhou4,wgao23,xliu96}@ncsu.edu`

## ABSTRACT

This work tackles a critical challenge in AI safety research under limited compute: given a fixed computation budget, how can one maximize the strength of iterative adversarial attacks? Coarsely reducing the number of attack iterations lowers cost but substantially weakens effectiveness. To fulfill the attainable attack efficacy within a constrained budget, we propose a fine-grained control mechanism that selectively recomputes layer activations across both iteration-wise and layer-wise levels. Extensive experiments show that our method consistently outperforms existing baselines at equal cost. Moreover, when integrated into adversarial training, it attains comparable performance with only 30% of the original budget.

## 1 INTRODUCTION

Adversarial attacks, which craft imperceptible perturbations to input data to degrade the performance of deep learning models, have become a central topic in the safety and robustness of AI systems. From the attack perspective, iterative adversarial methods such as Projected Gradient Descent (PGD) (Madry et al., 2017) are widely adopted as strong oracles to benchmark the robustness of modern neural networks. From the defense perspective, adversarial training (Madry et al., 2017; Zhang et al., 2019; Wang et al., 2019) has been validated as the most effective method, which critically depends on strong iterative attacks to generate challenging adversaries during training.

Despite their broad impact, iterative attacks are notoriously expensive because each step requires both a forward and backward pass. In robustness evaluation, PGD with 20, 50, or even 100 steps is typically used, which incurs roughly 40–200× the cost of a natural inference. In adversarial training, the standard PGD-10 procedure already costs around 10× more than natural training. As model architectures and datasets continue to grow, this heavy computational burden severely limits the scalability of both attacks and defenses for large-scale models. Normally, more computation in attacks (e.g., more iterative PGD steps) means stronger attack. This raises a critical question: **Given a prescribed computation budget, how can we maximize attack strength?**

This question is crucial in the era of increasingly large models and limited computational resources, especially for researchers and practitioners outside major industrial labs. To enhance attack strength and transferability, prior work has explored several complementary directions, including momentum-based updates (Dong et al., 2017; Lin et al., 2019), diverse input transformations (Xie et al., 2019), and translation-invariant gradients (Dong et al., 2019). Nevertheless, these approaches still incur substantial performance degradation when the iteration count is reduced and fail to sustain high attack efficacy under strict computational constraints.

We attribute this limitation to a common design choice: existing methods perform full-precision computation across all layers and treat iteration count as the sole, *coarse-grained* lever for controlling cost. From a combinatorial-optimization perspective, it is unlikely to be optimal to restrict computation allocation to a fixed number of iterations applied uniformly to every layer. As we observe in Section 3.2, during iterative attacks the activations across successive steps quickly become highly correlated, and the residual change rate varies across layers. This suggests that we should employ more *fine-grained* computation control at both the iteration-wise and layer-wise levels.

Motivated by this insight, we propose the *Spiking Iterative Attack*, an event-driven iterative scheme that adaptively executes fine-grained control over activation computation. Specifically, the spiking mechanism performs full-precision computation only when the relative activation change exceeds

a threshold; otherwise, the previous output is reused. To avoid gradient vanishing caused by naive reuse, we introduce a virtual surrogate gradient that preserves informative backward signals while retaining most of the forward-pass savings. Our main contributions can be summarized as follows:

- We identify and quantify substantial computation redundancy in multi-step iterative attacks, demonstrating that intermediate activations are highly correlated with strong layer-wise similarity.

- We introduce a combinatorial optimization perspective to reveal limitations of existing attacks: the coarse-grained computation allocation limits attack strength given a computation budget.

- We propose a *spiking forward computation* scheme to reduce redundant computation by adaptive computation. Moreover, a *virtual surrogate gradient* technique is proposed to preserve backward signals despite activation reuse. Together, the proposed attack algorithm (Spiking-PGD) ensures effective adversarial updates and enhances attack effectiveness under given computation budget.

- We empirically demonstrate that Spiking-PGD expands the efficiency–effectiveness Pareto frontier: it achieves comparable or superior attack success rates at significantly lower computational cost on both vision and graph benchmarks. Moreover, incorporating the spiking mechanism into adversarial training reduces training cost without degrading final clean or robust accuracy.

## 2    RELATED WORK

Adversarial attacks have become a central topic in the safety and robustness of AI systems, with a large body of work dedicated to improving their attack efficacy and efficiency. The Fast Gradient Sign Method (FGSM) (Goodfellow et al., 2014) introduced a simple one-step gradient update, while its iterative variant I-FGSM (Kurakin et al., 2018) achieved higher success rates by applying multiple small steps. Building on these foundations, several refinements have enhanced attack strength and transferability while requiring a small number of iterations. The Projected Gradient Descent (PGD) attack (Madry et al., 2017), often considered the canonical iterative variant of FGSM, adds a random start to the update rule. MI-FGSM (Dong et al., 2017) incorporates momentum to stabilize gradient updates, DI-FGSM (Xie et al., 2019) applies diverse input transformations to improve transferability, and extensions such as TI-FGSM (Dong et al., 2019) and NI-FGSM (Lin et al., 2019) further refine the optimization by introducing translation invariance and Nesterov momentum, respectively.

Beyond the white-box setting, efficiency has also been studied in black-box scenarios where query complexity is the dominant cost. Bandit-based methods (Ilyas et al., 2018) leverage gradient priors to reduce the number of queries, while decision-based attacks such as Boundary Attack (Brendel et al., 2017) and HopSkipJumpAttack (Chen et al., 2020) progressively refine adversarial examples using only hard-label feedback. More recently, Square Attack (Andriushchenko et al., 2020) demonstrated that simple random search over patchwise perturbations can achieve remarkable query efficiency. Sparse adversarial attacks, such as the One-Pixel Attack (Su et al., 2019) and SparseFool (Modas et al., 2019), further reduce perturbation cost by focusing on minimal, geometry-driven modifications.

In summary, prior research has explored efficiency either by reducing the number of gradient steps, minimizing queries, or constraining perturbations. In contrast, our proposed Spiking iterative attack seeks to improve efficiency from a complementary and orthogonal perspective: by exerting fine-grained control over activation computation, it selectively skips redundant operations during iterative updates while retaining the strong adversarial strength.

## 3    PRELIMINARY

In this section, we first present the necessary technical background on gradient-based iterative adversarial attacks and their associated computational overhead. We then conduct a preliminary study that reveals significant redundancy in the computations across attack iterations.

**Notations.** Consider an $L$-layer model, where $\mathcal{A}^{(l)}(\cdot)$ denotes the $l$-th layer in the model ($l \in \{1, \ldots, L\}$). The input and output activations of $\mathcal{A}^{(l)}$ are denoted by $\boldsymbol{a}^{(l)}$ and $\boldsymbol{o}^{(l)} = \mathcal{A}^{(l)}(\boldsymbol{a}^{(l)})$, respectively. Consider a $T$-step iterative attack that passes through the model $T$ times, then we have $\boldsymbol{o}_t^{(l)} = \mathcal{A}^{(l)}(\boldsymbol{a}_t^{(l)})$ at time step $t$ ($t \in \{1, \ldots, T\}$).

## 3.1 Expensive Computation in Iterative Adversarial Attacks.

Gradient-based iterative attacks (Madry et al., 2017; Kurakin et al., 2018; Dong et al., 2017; 2019) construct adversarial examples by iteratively perturbing the input to increase the adversarial loss. Although individual methods differ only in minor implementation details, they follow the general procedure summarized in Algorithm 1. Starting from an initial point $x_1$, it performs a forward propagation to evaluate the loss and a backward propagation to obtain the gradient with respect to input data $x$. The projected gradient update at step $t$ is the following: $x_{t+1} = \Pi_{\mathcal{B}_\epsilon(x)}(x_t + \alpha \nabla_x \mathcal{L}(x_t, y))$, where $\mathcal{L}$ denotes the adversarial loss (e.g., cross-entropy), $\alpha$ is the step size, and $\Pi_{\mathcal{B}_\epsilon(x)}$ projects

---

**Algorithm 1** Gradient-based Adversarial Attack

**Require:** clean input $x$, label $y$, adversarial loss $\mathcal{L}$, attack budget $\epsilon$, step size $\alpha$, iteration $T$.
**Ensure:** adversarial example $\tilde{x}$
1: $x_1 \leftarrow x$             ▷ initialization
2: **for** $t = 1, \ldots, T$ **do**
3:     $x_t \to o_t^{(1)} \cdots o_t^{(l)} \cdots \to o_t^{(L)} \to \mathcal{L}$
4:     $\frac{\partial \mathcal{L}}{\partial x_t} \leftarrow \frac{\partial \mathcal{L}}{\partial o_t^{(1)}} \cdots \frac{\partial \mathcal{L}}{\partial o_t^{(l)}} \cdots \leftarrow \frac{\partial \mathcal{L}}{\partial o_t^{(L)}} \leftarrow \mathcal{L}$
5:     $x_{t+1} \leftarrow \Pi_{\mathcal{B}_\epsilon(x)}\left(x_t + \alpha \cdot \frac{\partial \mathcal{L}}{\partial x_t}\right)$
6: **end for**
7: **return** $\tilde{x} \leftarrow x_{T+1}$

---

the perturbed data back onto the allowed perturbation set $\mathcal{B}_\epsilon(x)$ given the attack budget $\epsilon$.

With $T$ iterations, iterative attacks produce much stronger adversaries than single-step methods, at the cost of $T$ forward and backward passes. Although iterative adversarial attacks are among the most effective methods for evaluating model robustness, they are also extremely expensive to perform multiple gradient calculations per input, with each step involving both a forward and backward pass through the network. Furthermore, when these attacks are incorporated into adversarial training, the cost compounds, as each training step must generate new adversarial examples through repeated iterations, which limits its applicability in real-world large-scale deployments.

## 3.2 Computation Redundancy in Iterative Adversarial Attacks.

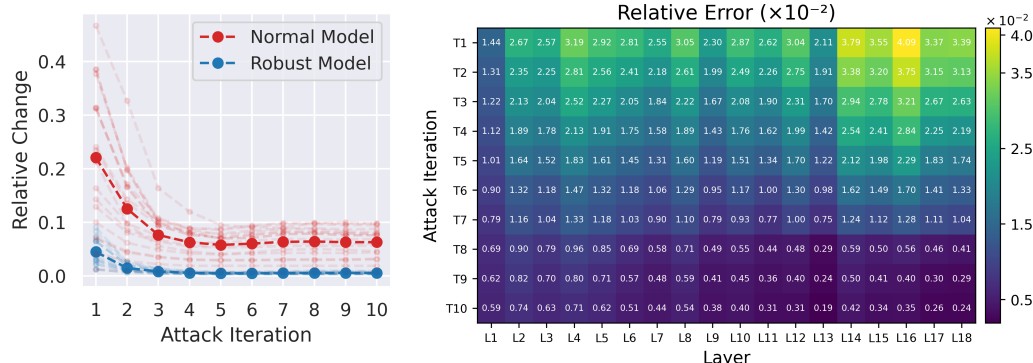

Figure 1: Activation relative change $\|a_t - a_{t-1}\| / \|a_t\|$ for ResNet-18 on CIFAR-10. **Left**: per-layer curves (light color) and layer average (dark color) for normally trained (red) and adversarially trained (blue) models. **Right**: heatmap of relative change across layers and attack iterations.

As shown in Algorithm 1, iterative attacks typically update the adversarial perturbation via projected gradient steps until convergence. As a result, the perturbed input and the network's hidden activations change smoothly, especially in the later stages when the optimizer has largely stabilized. Motivated by this intuition, we conduct a preliminary study on CIFAR-10 (Krizhevsky et al., 2009) with ResNet-18 (He et al., 2016) to validate our argument. To be specific, we measure the relative activation change $\|a_t^{(l)} - a_{t-1}^{(l)}\| / \|a_t^{(l)}\|$ between consecutive attack iterations $t - 1$ and $t$ for each layer $l$. We refer to a standard model trained without adversarial examples as the "Normal Model", and to an adversarially trained model as the "Robust Model". Figure 1 (Left) visualizes the relative activation change per layer using light-colored lines, while the dark lines represent the layer-wise average. Additionally, a heat map in Figure 1 (Right) is used to illustrate the distribution of relative changes across iterations and layers. We can make the following observations from the results: **(1)** Figure 1 (Left) shows that the activations $\{a_t^{(l)}\}_{t=1}^T$ become highly similar after a small number of iterations, i.e., $\|a_t^{(l)} - a_{t-1}^{(l)}\| / \|a_t^{(l)}\|$ decreases rapidly. This indicates substantial redundancy in repeated computations across attack iterations. Additionally, the "Robust Model" exhibits markedly smaller activation changes than the "Normal Model"; **(2)** Figure 1 (Right) shows that while all layers

follow the same overall decreasing trend, the decay rate of relative activation change varies across layer. These findings motivate *iteration-wise* and *layer-wise* fine-grained control of computation.

## 4 SPIKING ITERATIVE ATTACK

In this section, we first formulate the combinatorial optimization problem underlying activation computation in iterative attacks and highlight the limitations of existing approaches, which impose coarse-grained constraints on the search space (Section 4.1). We then introduce a novel and efficient Spiking Iterative Attack that leverages fine-grained spiking-based forward computation (Section 4.2) in conjunction with a faithful virtual surrogate gradient for the backward computation (Section 4.3).

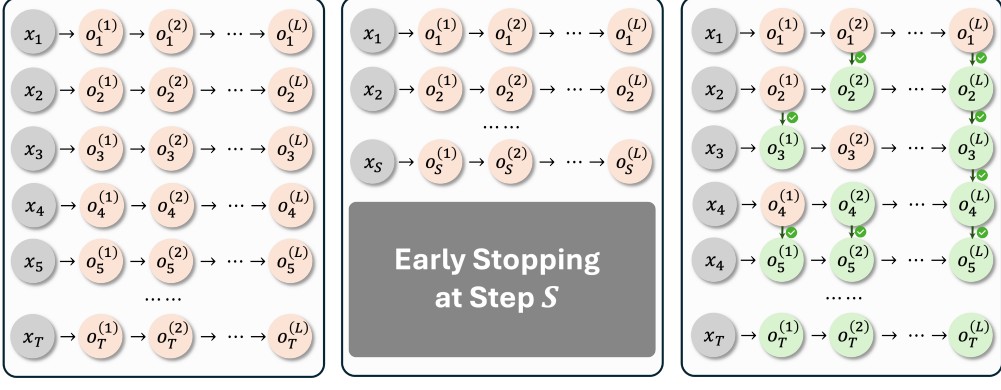

(a) Vanilla Iterative Attack      (b) Coarse-Grained Attack      (c) Fine-Grained Attack

Figure 2: Three attack execution patterns: (a) Vanilla iterative attack: every layer $l \in [L]$ is fully computed at every attack iteration $t \in [T]$; (b) Coarse-grained attack: all layers are computed for $t \leq S$ and skipped for $t > S$; and (c) Fine-grained attack: selectively controls the computation scheme to maximize attack strength under a given computational budget.

### 4.1 ITERATIVE ATTACK AS COMBINATORIAL OPTIMIZATION UNDER LIMITED BUDGET

Multi-step iterative adversarial attacks are costly because each attack step typically performs a full forward and backward pass through the network. Let $C_t$ denote the computational cost at iteration $t$, then the total cost scales as $\sum_{t=1}^{T} C_t$. Under a strict computation budget $C_{\text{total}}$, a common strategy is to reduce the iteration count in iterative attacks to some $S \leq T$ (early stopping; see Figure 2b), which leads to the following *coarse-grained* problem:

$$\max_{S \in \{1,...,T\}} \mathcal{L}\big(\boldsymbol{x}_{S+1}, y\big) \quad \text{s.t.} \quad \sum_{t=1}^{S} C_t \leq C_{\text{total}}, \tag{1}$$

where $\boldsymbol{x}_{S+1}$ denotes the perturbed input after $S$ attack full iterations. This early-stop constraint enforces a block-structured computation pattern, which severely restricts the admissible computation schedules, making the coarse-grained solution tend to be suboptimal.

We instead consider a fine-grained control over which layer computations are executed at each iteration (Figure 2 (c)). Decomposing the per-iteration cost by layer, we write $C_t = \sum_{l=1}^{L} C_{t,l}$. Let $\Delta = (\delta_{t,l}) \in \{0,1\}^{T \times L}$ be a binary mask where $\delta_{t,l} = 1$ indicates that layer $l$ is fully computed at attack iteration $t$, while $\delta_{t,l} = 0$ indicates reuse of previously computed activations. The corresponding *fine-grained* optimization is the following:

$$\max_{\Delta \in \{0,1\}^{T \times L}} \mathcal{L}\big(\boldsymbol{x}_{T+1}(\Delta), y\big) \quad \text{s.t.} \quad \sum_{t=1}^{T} \sum_{l=1}^{L} C_{t,l}\, \delta_{t,l} \leq C_{\text{total}}, \tag{2}$$

where $\boldsymbol{x}_{T+1}(\Delta)$ denotes the final perturbed input after $T$ attack steps executed under mask $\Delta$. The following proposition formalizes the relationship between the two formulations: the coarse-grained problem Eq. (1) is a subproblem of the fine-grained problem Eq. (2).

**Proposition 4.1.** *Let $V_{\text{coarse}}$ and $V_{\text{fine}}$ denote the optimal objective values of Eq. (1) and Eq. (2), respectively, then we have: $V_{\text{coarse}} \leq V_{\text{fine}}$. The feasible set of Eq. (1) can be embedded into the feasible set of Eq. (2) by masks that compute layers for iterations $t \leq S$ and skip layers for $t > S$.*

*Remark* 4.2. *Proposition 4.1 shows that early-stopping is a block-structured subset of the more expressive layer-wise masking decisions. The fine-grained formulation can therefore improve attack strength under the same budget, at the expense of a larger combinatorial search space.*

So far we have introduced a fine-grained, layer-wise optimization as an alternative to coarse-grained early stopping. Despite its intuitive appeal, solving the fine-grained problem in Eq. (2) is challenging for the following reasons: **(1)** Vast search space: The binary mask $\Delta \in \{0,1\}^{T \times L}$ induces a combinatorial search space of size $\mathcal{O}(2^{T \times L})$ lead to exhaustive search; **(2)** Expensive objective evaluations: Each evaluation of the attack objective $\mathcal{L}(\mathbf{x}_{T+1}(\Delta), y)$ requires simulating the entire masked forward/backward computation under $\Delta$, eliminates the practicality of classic optimization methods; and **(3)** Broken gradient flow: Reusing cached activations breaks the usual computation graph, leading to vanishing gradient with respect to the input perturbation (See Section 4.3).

To address these issues, we propose a novel *spiking forward computation* scheme (Section 4.2) that enables fine-grained control of per-layer computation based on our preliminary study of redundant computation during iterative attacks. To mitigate the resulting gradient-breakage, we introduce a *virtual surrogate gradient* (Section 4.3) that preserves gradient information during back-propagation.

## 4.2 SPIKING FORWARD COMPUTATION

To solve the combinatorial optimization in Eq. (2), one may naturally consider a range of classical approaches such as brute-force enumeration, greedy heuristics, dynamic programming (Bellman, 1957), continuous relaxations (Raghavan & Thompson, 1987), or a variety of meta-heuristic algorithms (Holland, 1975; Koza, 1992; Hansen & Ostermeier, 2001; Kirkpatrick et al., 1983). However, in our setting they become impractical: the decision variable $\Delta \in \{0,1\}^{T \times L}$ induces an exponentially large search space, and each candidate mask requires an expensive end-to-end attack simulation to evaluate $\mathcal{L}(\boldsymbol{x}_{T+1}(\Delta), y)$. Motivated by the empirical redundancy observed in Section 3.2, we propose an efficient *spiking forward computation* scheme that enforces layer-wise recomputation via a single tunable threshold parameter $\rho \in [0,1]$. This threshold controls a lightweight per-layer selector to enable strong attack efficacy while meeting a prescribed computational budget.

Consider a linear mapping $\mathcal{A}^{(l)}$ in the model, with sequential inputs $\{\boldsymbol{a}_t^{(l)}\}_{t=1}^T$ and outputs $\{\boldsymbol{o}_t^{(l)}\}_{t=1}^T$, where $\boldsymbol{o}_t^{(l)} = \mathcal{A}^{(l)}(\boldsymbol{a}_t^{(l)})$. Exploiting the linearity of $\mathcal{A}^{(l)}$, vanilla forward computation can be reformulated as shown in Figure 3 (left), where each output is expressed as the accumulation of the previous output plus the residual between consecutive activations. This view naturally suggests that many computations are redundant when residuals are small.

$$
\begin{cases}
\boldsymbol{o}_1^{(l)} = \mathcal{A}^{(l)}(\boldsymbol{a}_1^{(l)}) \\
\boldsymbol{o}_2^{(l)} = \mathcal{A}^{(l)}(\boldsymbol{a}_2^{(l)}) = \mathcal{A}^{(l)}(\boldsymbol{a}_2^{(l)} - \boldsymbol{a}_1^{(l)}) + \boldsymbol{o}_1^{(l)} \\
\cdots \\
\boldsymbol{o}_t^{(l)} = \mathcal{A}^{(l)}(\boldsymbol{a}_t^{(l)}) = \mathcal{A}^{(l)}(\boldsymbol{a}_t^{(l)} - \boldsymbol{a}_{t-1}^{(l)}) + \boldsymbol{o}_{t-1}^{(l)} \\
\cdots \\
\boldsymbol{o}_T^{(l)} = \mathcal{A}^{(l)}(\boldsymbol{a}_T^{(l)}) = \mathcal{A}^{(l)}(\boldsymbol{a}_T^{(l)} - \boldsymbol{a}_{T-1}^{(l)}) + \boldsymbol{o}_{T-1}^{(l)}
\end{cases}
\Rightarrow
\begin{cases}
\boldsymbol{o}_1^{(l)} = \hat{\boldsymbol{o}}_1^{(l)} = \mathcal{A}^{(l)}(\boldsymbol{a}_1^{(l)}) \\
\boldsymbol{o}_2^{(l)} \approx \hat{\boldsymbol{o}}_2^{(l)} = \mathcal{A}^{(l)}(\mathcal{S}_\rho(\boldsymbol{a}_2^{(l)}, \boldsymbol{a}_1^{(l)})) + \hat{\boldsymbol{o}}_1^{(l)} \\
\cdots \\
\boldsymbol{o}_t^{(l)} \approx \hat{\boldsymbol{o}}_t^{(l)} = \mathcal{A}^{(l)}(\mathcal{S}_\rho(\boldsymbol{a}_t^{(l)}, \boldsymbol{a}_{t-1}^{(l)})) + \hat{\boldsymbol{o}}_{t-1}^{(l)} \\
\cdots \\
\boldsymbol{o}_T^{(l)} \approx \hat{\boldsymbol{o}}_T^{(l)} = \mathcal{A}^{(l)}(\mathcal{S}_\rho(\boldsymbol{a}_T^{(l)}, \boldsymbol{a}_{T-1}^{(l)})) + \hat{\boldsymbol{o}}_{T-1}^{(l)}
\end{cases}
$$

Figure 3: Comparison of vanilla forward computation and spiking forward computation for adversarial attacks. **Left**: In vanilla forward computation, the current output is composed of the previous output and the activation residual, leveraging the linearity of $\mathcal{A}^{(l)}$. **Right**: Spiking forward computation applies a spiking function $\mathcal{S}_\rho$ to the residual to selectively update the activations.

Motivated by empirical observations in Section 3.2 that activations across attack iterations are highly correlated, we introduce a spiking mechanism to determine whether an update should be triggered:

$$
S_\rho(\boldsymbol{a}_t, \boldsymbol{a}_{t-1}) = \begin{cases}
\mathbf{1} \cdot (\boldsymbol{a}_t - \boldsymbol{a}_{t-1}), & \|\boldsymbol{a}_t - \boldsymbol{a}_{t-1}\| / \|\boldsymbol{a}_t\| \geq \rho, \\
\mathbf{0} \cdot (\boldsymbol{a}_t - \boldsymbol{a}_{t-1}), & \|\boldsymbol{a}_t - \boldsymbol{a}_{t-1}\| / \|\boldsymbol{a}_t\| < \rho,
\end{cases}
\tag{3}
$$

where $\rho$ is a pre-defined threshold controlling the sensitivity of the spiking gate. Intuitively, the gate "fires" only when the relative change in activation is sufficiently large, thereby avoiding unnecessary recomputation for negligible updates.

This spiking formulation leads to two distinct cases: **(1)** If $\|\boldsymbol{a}_t - \boldsymbol{a}_{t-1}\| / \|\boldsymbol{a}_t\|$ is larger than $\rho$, we compute a new output: $\hat{\boldsymbol{o}}_t^{(l)} = \mathcal{A}^{(l)}(\boldsymbol{a}_t^{(l)} - \boldsymbol{a}_{t-1}^{(l)}) + \boldsymbol{o}_{t-1}^{(l)} = \mathcal{A}^{(l)}(\boldsymbol{a}_t^{(l)})$; and **(2)** if $\|\boldsymbol{a}_t - \boldsymbol{a}_{t-1}\| / \|\boldsymbol{a}_t\|$

is smaller than $\rho$, the update degenerates to: $\hat{\boldsymbol{o}}_t^{(l)} = \mathcal{A}^{(l)}(\boldsymbol{0}) + \boldsymbol{o}_{t-1}^{(l)} = \boldsymbol{o}_{t-1}^{(l)}$, reusing the previous output and skipping redundant computation. Through this mechanism, the forward pass becomes event-driven rather than iteration-driven, leading to substantial savings in computation. Importantly, the spiking threshold $\rho$ allows explicit control over the trade-off between efficiency and efficacy, enabling the attack to flexibly adapt to different resource budgets.

## 4.3 VIRTUAL BACKWARD COMPUTATION

(a) Exact Gradient      (b) Vanishing Gradient      (c) Virtual Gradient

Figure 4: Gradient computation mechanisms. (a) Exact gradient $\frac{\partial \mathcal{L}}{\partial \boldsymbol{a}_t} = \mathcal{A}^{\top}\left(\frac{\partial \mathcal{L}}{\partial \hat{\boldsymbol{o}}_t}\right)$: when the layer output $\hat{o}_t$ is freshly computed, the gradient w.r.t. $\boldsymbol{a}_t$ follows the standard chain rule. (b) Vanishing gradient $\frac{\partial \mathcal{L}}{\partial \boldsymbol{a}_t} = \boldsymbol{0} \cdot \frac{\partial \mathcal{L}}{\partial \hat{\boldsymbol{o}}_t} = \boldsymbol{0}$: when $\hat{o}_t$ is reused from $\hat{o}_{t-1}$, the spiking gate blocks the path to $\boldsymbol{a}_t$, yielding zero gradient. (c) Virtual gradient $\frac{\partial \mathcal{L}}{\partial \boldsymbol{a}_t} = \mathcal{A}^{\top}\left(\frac{\partial \mathcal{L}}{\partial \hat{\boldsymbol{o}}_{t-1}}\right)$: our virtual surrogate gradient manually restores the backward path by manually applying $\mathcal{A}^{\top}$ to the upstream gradient.

**Gradient Vanishing.** When computing backward gradient in the spiking computation described in Section 4.2, the default `torch.autograd` backward pass follows the chain rule through the spiking gate. For a linear layer $\mathcal{A}^{(l)}$, we obtain

$$\frac{\partial \mathcal{L}}{\partial \boldsymbol{a}_t^{(l)}} = \mathcal{S}_\rho' \cdot \mathcal{A}^{(l)\top}\left(\frac{\partial \mathcal{L}}{\partial \hat{\boldsymbol{o}}_t^{(l)}}\right) = \begin{cases} \mathcal{A}^{(l)\top}\left(\frac{\partial \mathcal{L}}{\partial \hat{\boldsymbol{o}}_t^{(l)}}\right), & \|\boldsymbol{a}_t - \boldsymbol{a}_{t-1}\|/\|\boldsymbol{a}_t\| \geq \rho, \Rightarrow \text{ Figure 4 } (a) \\ \boldsymbol{0}, & \|\boldsymbol{a}_t - \boldsymbol{a}_{t-1}\|/\|\boldsymbol{a}_t\| < \rho, \Rightarrow \text{ Figure 4 } (b) \end{cases}$$

where $S_\rho'$ is the derivative of the spiking mask. The second case corresponds to Figure 4 (b): because the activation is reused, the forward value $\hat{\boldsymbol{o}}_t^{(l)}$ is set to the cached $\hat{\boldsymbol{o}}_{t-1}^{(l)}$, and the `autograd` engine produces a zero gradient to $\boldsymbol{a}_t^{(l)}$. Practically, this results in *gradient vanishing* for $\boldsymbol{a}_t^{(l)}$ and thus prevents the input perturbation $\boldsymbol{x}$ from receiving a useful update via this layer.

**Virtual Surrogate Gradient.** To address this problem, we introduce a *virtual surrogate gradient* that restores a more faithful backward signal when the spiking gate suppresses the standard gradient. Conceptually (Figure 4 (c)), we manually inject an approximate backward mapping from the reused output $\hat{o}_{t-1}$ back to the current input $\boldsymbol{a}_t$, replacing the zero that `autograd` would otherwise propagate. Concretely, our implementation proceeds in two steps: (1) **Step One**: We mark the cached output $\hat{\boldsymbol{o}}_{t-1}^{(l)}$ as requiring gradients via `requires_grad_()` so that we can capture $\partial \mathcal{L}/\partial \hat{\boldsymbol{o}}_{t-1}^{(l)}$ and store it during the backward pass; (2) **Step Two**: When the layer activation is reused, we retrieve the cached upstream gradient $\partial \mathcal{L}/\partial \hat{\boldsymbol{o}}_{t-1}^{(l)}$ and manually compute the surrogate gradient $\mathcal{A}^{(l)\top}\left(\partial \mathcal{L}/\partial \hat{\boldsymbol{o}}_{t-1}^{(l)}\right)$ by placing the $\mathcal{A}^{(l)\top}$ between $\hat{\boldsymbol{o}}_{t-1}^{(l)}$ and $\boldsymbol{a}_t^{(l)}$ in the computational graph via `register_hook`. In practice, this is implemented efficiently using the corresponding low-level gradient primitives (e.g., `torch.nn.grad.conv2d_input` for convolutional layers, or a matrix-multiplication for dense layers).

With the surrogate gradient in place, the backward rule becomes a piecewise function: if a layer is fully computed, we rely on the true backward provided by `autograd`; if it is reused, we substitute the surrogate gradient via `register_hook`. This hybrid rule restores meaningful gradient flow to the input while preserving the forward-pass savings. An overview of the full Spiking Iterative Attack with PGD is given in Algorithm 2.

## 5 EXPERIMENT

In this section, we first validate the effectiveness of the proposed Spiking Iterative Attack in vision and graph domains. We then incorporate Spiking-PGD into adversarial training to substantially improve efficiency without compromising final performance. Finally, we conduct several ablation studies to analyze the underlying mechanisms and behaviors of Spiking-PGD.

---

**Algorithm 2** Spiking-PGD

---

**Require:** clean input $x$, label $y$, adversarial loss $\mathcal{L}$, attack budget $\epsilon$, step size $\alpha$, iterations $T$.
**Ensure:** adversarial example $\tilde{x}$

1: $x_1 \leftarrow x + \delta$      ▷ optional random start $\delta$ s.t. $\|\delta\|_p \leq \epsilon$; else $\delta \leftarrow 0$
2: **for** $t = 1, \ldots, T$ **do**
3:      **for** $l = 1$ to $L$ **do**
4:          $\hat{o}_t^{(l)} = \begin{cases} o_{t-1}^{(l)}, & \text{if } t > 1 \text{ and } \frac{\|a_t^{(l)} - a_{t-1}^{(l)}\|}{\|a_t^{(l)}\|} \leq \rho. \\ \mathcal{A}^{(l)}(a_t^{(l)}), & \text{if } t = 1 \text{ or } \frac{\|a_t^{(l)} - a_{t-1}^{(l)}\|}{\|a_t^{(l)}\|} \geq \rho. \end{cases}$      ▷ forward propagation
5:      **end for**
6:      **for** $l = L$ to $1$ **do**
7:          $\frac{\partial \mathcal{L}}{\partial a_t^{(l)}} = \begin{cases} \mathcal{A}^{(l)\top}\left(\frac{\partial \mathcal{L}}{\partial \hat{o}_{t-1}^{(l)}}\right), & \text{if } t > 1 \text{ and } \frac{\|a_t^{(l)} - a_{t-1}^{(l)}\|}{\|a_t^{(l)}\|} \leq \rho. \\ \mathcal{A}^{(l)\top}\left(\frac{\partial \mathcal{L}}{\partial \hat{o}_t^{(l)}}\right), & \text{if } t = 1 \text{ or } \frac{\|a_t^{(l)} - a_{t-1}^{(l)}\|}{\|a_t^{(l)}\|} \geq \rho. \end{cases}$      ▷ backward propagation
8:      **end for**
9:      $x_{t+1} \leftarrow \Pi_{\mathcal{B}_\epsilon(x)}\left(x_t + \alpha \cdot \text{sign}\left(\frac{\partial \mathcal{L}}{\partial x_t}\right)\right)$      ▷ adversary update
10: **end for**
11: **return** $\tilde{x} \leftarrow x_{T+1}$

---

### 5.1 EXPERIMENTAL SETTINGS

**Datasets & Backbone Models.** We conduct the experiments on several datasets: for vision task, we use CIFAR10 (Krizhevsky et al., 2009), CIFAR100 (Krizhevsky et al., 2009) and Tiny-ImageNet (Le & Yang, 2015); for graph task, we use Cora and Citeseer (Sen et al., 2008). For backbone models, we select ResNet18 (He et al., 2016) for vision task and GCN (Kipf, 2016) for graph task.

**Baseline Attacks.** For the vision task, we evaluate iterative attacks including PGD (Madry et al., 2017), I-FGSM (Kurakin et al., 2018), and MI-FGSM (Dong et al., 2017). For the graph task, we evaluate the PGD topology attack (Xu et al., 2019). We follow all the settings in the original papers except for the number of attack iterations $T$.

**Computation Cost.** We set the reference number of iterations to $T_0 = 20$ for vision tasks and $T_0 = 200$ for graph tasks. For baseline iterative attacks with $T$ iterations, the relative computational cost is defined as $T/T_0 \times 100\%$. For our Spiking-PGD, the relative computational cost is measured by the proportion of full-precision operations executed over the entire iterative attack in the model.

### 5.2 ADVERSARIAL ATTACK STRENGTH

To evaluate the strength of different adversarial attacks under any prescribed budget, we report model accuracy under attack across a range of relative computational costs (see Section 5.1 for details on cost measurement). For baseline iterative attacks, we vary the number of iterations $T$ to span low- to high-budget regimes, corresponding to relative costs of $T/T_0 \times 100\%$. For our Spiking-PGD, we adjust the spiking threshold $\rho \in [0, 1]$ to obtain operating points corresponding to different proportions of full updates. We conduct our evaluation on vision tasks (CIFAR-10, CIFAR-100, and Tiny-ImageNet) and graph tasks (Cora and Citeseer).

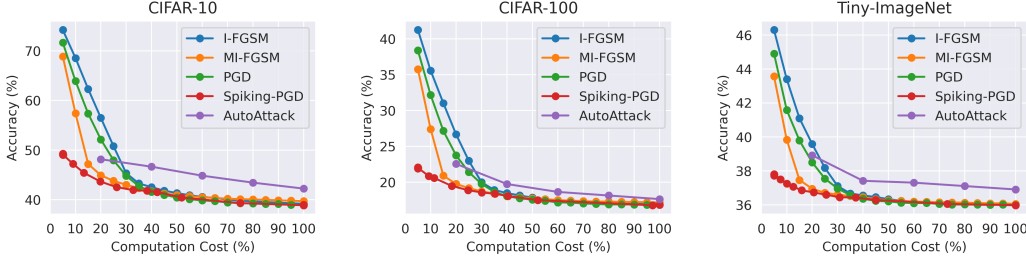

Figure 5: Comparison of model accuracy under attack versus computation cost on CIFAR-10, CIFAR-100, and Tiny-ImageNet. Spiking-PGD consistently achieves better attack strength than baseline iterative attacks (I-FGSM, MI-FGSM, PGD, AutoAttack), with the performance gap most pronounced in the low-computation regime.

**Pixel Perturbation on Images.** Figure 5 presents a comparison of attack strength with various computation costs for our Spiking-PGD and several baseline methods across three vision benchmarks: CIFAR-10, CIFAR-100, and Tiny-ImageNet. Several key observations can be made:

- Across all datasets, for any given computation cost, Spiking-PGD consistently achieves higher attack strength than baseline iterative attacks. The performance gap is especially pronounced in the low-budget regime, highlighting that Spiking-PGD utilizes computation more efficiently. This advantage stems from its fine-grained control over computation via the spiking threshold, in contrast to the coarse-grained iteration-based control used by conventional methods.

- Among baseline attacks, MI-FGSM outperforms I-FGSM and PGD due to the use of momentum in gradient updates to stabilizes optimization. However, when the number of iterations is reduced to lower computation cost, MI-FGSM suffers a significant drop in attack strength. This indicates that iteration reduction alone cannot preserve effectiveness, leaving considerable room for improvement—an efficiency gap that Spiking-PGD bridges through fine-grained computation allocation. Additionally, although AutoAttack offers strong attack strength, it does so at a substantial computational cost. Under an equal computation budget, however, AutoAttack performs worse than the other attacks.

**Structure Attack on Graphs.** Figure 6 shows results on two citation network benchmarks (Cora, Citeseer) for PGD topology attacks and our Spiking-PGD. We can make similar observations as vision task: (1) PGD attack strength falls sharply when we reduce computational effort. (2) The Spiking-PGD achieves noticeably higher attack strength than PGD across most budgets. The gap is particularly clear at small-to-moderate budgets, indicating that reusing computed quantities in graph updates preserves enough signal to guide structure perturbations effectively.

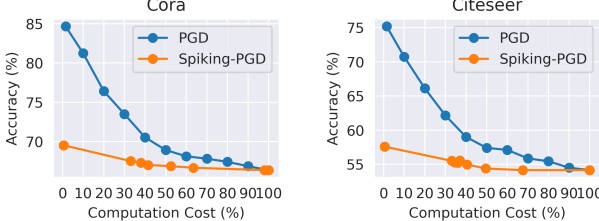

Figure 6: Comparison of model accuracy under attack versus computation cost on graph datasets (Cora and Citeseer). Spiking-PGD consistently achieves stronger attack performance than PGD across all computation budgets, with a particularly large advantage in the low-cost regime.

## 5.3 EFFICIENT ADVERSARIAL TRAINING

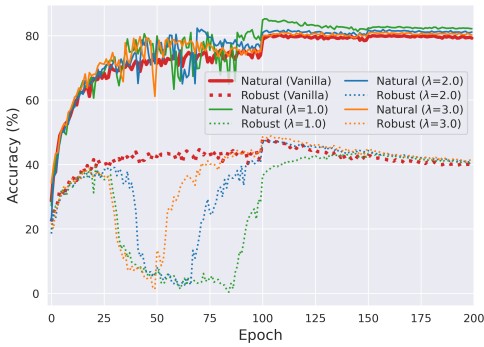
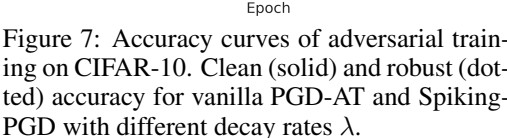
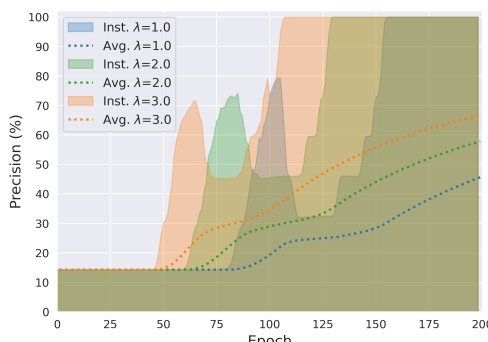

Figure 7: Accuracy curves of adversarial training on CIFAR-10. Clean (solid) and robust (dotted) accuracy for vanilla PGD-AT and Spiking-PGD with different decay rates $\lambda$.

Figure 8: Precision curves of adversarial training on CIFAR-10. Instantaneous (shaded) and average (dotted) precision for Spiking-PGD with different decay rates $\lambda$.

Beyond evaluating iterative attacks, we also integrate Spiking-PGD into adversarial training to reduce the cost of generating perturbations. In this setting, we investigate two scheduling strategies for controlling the spiking threshold $\rho$: a constant threshold and an exponential decay schedule. For the latter, the threshold at epoch $t$ is defined as $\rho(t) = \rho_0 \cdot \frac{e^{-\lambda t/N} - e^{-\lambda}}{1 - e^{-\lambda}}$, where $\rho_0 = 0.1$ is the initial threshold, $N = 200$ is the total number of epochs, and $\lambda$ is a hyperparameter that controls the decay rate. By construction, $\rho(0) = \rho_0$ and $\rho(T) = 0$. Intuitively, this schedule produces weaker (and cheaper) perturbations in the early stages of training, while progressively increasing attack strength and precision as training proceeds. This gradual refinement allows the model to stabilize early while still converging to a robust solution.

To better understand the training dynamics, we visualize how accuracy and precision evolve across epochs during adversarial training with Spiking-PGD. For accuracy curves in Figure 7, we track both clean and robust accuracy throughout training. For precision curves in Figure 8, we distinguish between the instantaneous precision at each epoch and the cumulative average precision over time. From the results, we can make key observations as follows: **(1)** Between epochs 25–50, robust accuracy drops sharply as low thresholds cause frequent reuses. As $\rho$ decays and more full computations are introduced, robust accuracy recovers and aligns with PGD-AT, showing that adversarial training can correct early inaccuracies once stronger perturbations appear. **(2)** The decay parameter $\lambda$ controls how quickly $\rho$ approaches zero. Smaller $\lambda$ (e.g., 1.0) prolongs efficiency but delays robustness recovery, while larger $\lambda$ (e.g., 3.0) accelerates refinement, yielding smoother curves closer to PGD-AT. **(3)** With $\lambda = 2.0$, Spiking-PGD nearly matches PGD-AT in best clean and robust accuracy while using under 30% of the computation.

## 5.4 Ablation Study

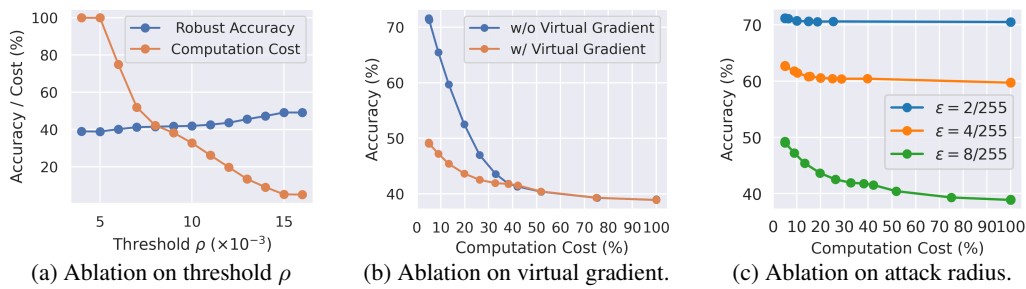

(a) Ablation on threshold $\rho$.     (b) Ablation on virtual gradient.     (c) Ablation on attack radius.

Figure 9: Ablation studies of Spiking-PGD. (a) Effect of spiking threshold $\rho$ on robust accuracy and computation cost. (b) Impact of the virtual surrogate gradient. (c) Effect of attack radius $\epsilon \in (2/255, 4/255, 8/255)$, where smaller radii lead to slower degradation in accuracy.

**Threshold $\rho$.** Figure 9a illustrates the sensitivity of attack performance to the spiking threshold $\rho$. As $\rho$ increases, computation cost decreases substantially since more operations are skipped, while attack strength decreases only moderately. This trade-off indicates that a higher threshold yields significant efficiency gains at the cost of a slight reduction in attack effectiveness.

**Virtual Surrogate Gradient.** To validate the effectiveness of our virtual surrogate gradient, we conduct an ablation study shown in Figure 9b. With the virtual surrogate gradient, attack strength improves substantially, especially under low computation cost. This demonstrates that the virtual surrogate gradient provides a faithful signal that enables stronger adversarial perturbations when computational resources are limited.

**Attack Radius $\epsilon$.** To examine the effect of the attack radius on Spiking-PGD, we evaluate three settings: $\epsilon = 2/255$, $4/255$, and $8/255$. From Figure 9c, we observe that Spiking-PGD yields greater efficiency improvements under smaller radii. With a small radius, attack strength decreases more gradually as computation cost is reduced, since the induced activation changes are less pronounced compared to larger radii.

## 6 Conclusion

Adversarial attacks have become a cornerstone of AI safety research, serving as a reliable means for robustness evaluation and system safeguarding. However, iterative attacks introduce substantial computational overhead, which limits both academic research and industrial deployment in the fields of AI safety and robustness. While reducing the number of attack iterations can significantly cut costs, it comes at the expense of attack strength. In this work, we propose spiking iterative attacks, which employ fine-grained forward computation to reduce overhead and a virtual surrogate gradient to preserve backward signals during activation reuse. Our experiments demonstrate the effectiveness of this method across various baselines under constrained budgets. Moreover, when integrated into adversarial training, our approach reduces computational cost by up to 70% without compromising final performance. In addition, the method naturally extends to black-box attacks that rely on iterative gradient estimation, since these attacks also produce temporally correlated queries that benefit from activation reuse. Overall, this method offers an orthogonal and complementary perspective to existing techniques, expanding the efficiency–effectiveness Pareto frontier and carrying crucial implications for the advancement of AI safety.

## ETHICS STATEMENT

This paper investigates efficient attack methods, which are valuable for evaluating and enhancing model robustness. However, we acknowledge the potential risk that such insights could be misused to target existing secure models. Beyond this, we have not identified any ethical concerns related to human subjects, data release practices, conflicts of interest or sponsorship, discrimination, bias or fairness, or issues of research integrity.

## REPRODUCIBILITY STATEMENT

We provide comprehensive details to facilitate the reproduction of our experiments. Specifically, the datasets, models, and attack methods are described in Section 5.1, along with the hyperparameters used in our proposed method. The code will be released upon paper acceptance.

## ACKNOWLEDGMENT

Weizhi Gao, Zhichao Hou, and Xiaorui Liu are supported by the National Science Foundation (NSF) under grant number IIS-2443182.

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

## A  REDUNDANCY IN ITERATIVE ATTACK.

To reveal the redundancy present in iterative attacks, we report the relative changes of activations and gradients in Table 1 and Table 2, respectively. The results show that both activations and gradients quickly converge to highly similar values after only a few iterations. This indicates that repeatedly recomputing them across all iterations is computationally wasteful. Moreover, the decay rates are not uniform across layers—some layers stabilize much faster than others. These observations highlight the need for fine-grained computation control, both at the iteration-wise and layer-wise levels, in order to effectively reduce redundancy while preserving attack strength.

| Normal Model | | | | | | | | | | | | | | | | | |
|---|---|---|---|---|---|---|---|---|---|---|---|---|---|---|---|---|---|
| Iteration \ Layer | 1 | 2 | 3 | 4 | 5 | 6 | 7 | 8 | 9 | 10 | 11 | 12 | 13 | 14 | 15 | 16 | 17 | 18 |
| 0 | 0.012 | 0.042 | 0.072 | 0.069 | 0.129 | 0.100 | 0.143 | 0.164 | 0.241 | 0.235 | 0.312 | 0.314 | 0.213 | 0.314 | 0.377 | 0.385 | 0.467 | 0.385 |
| 1 | 0.009 | 0.028 | 0.046 | 0.044 | 0.081 | 0.062 | 0.086 | 0.098 | 0.143 | 0.136 | 0.171 | 0.166 | 0.098 | 0.166 | 0.202 | 0.198 | 0.327 | 0.198 |
| 2 | 0.008 | 0.022 | 0.036 | 0.035 | 0.064 | 0.049 | 0.068 | 0.075 | 0.104 | 0.096 | 0.108 | 0.094 | 0.056 | 0.094 | 0.105 | 0.100 | 0.164 | 0.100 |
| 3 | 0.007 | 0.019 | 0.031 | 0.030 | 0.055 | 0.042 | 0.057 | 0.063 | 0.089 | 0.081 | 0.089 | 0.077 | 0.046 | 0.077 | 0.087 | 0.081 | 0.120 | 0.081 |
| 4 | 0.007 | 0.019 | 0.030 | 0.029 | 0.053 | 0.040 | 0.055 | 0.062 | 0.088 | 0.079 | 0.086 | 0.072 | 0.041 | 0.071 | 0.077 | 0.070 | 0.095 | 0.070 |
| 5 | 0.007 | 0.019 | 0.030 | 0.029 | 0.054 | 0.042 | 0.058 | 0.066 | 0.094 | 0.085 | 0.092 | 0.077 | 0.044 | 0.076 | 0.081 | 0.071 | 0.091 | 0.071 |
| 6 | 0.007 | 0.019 | 0.031 | 0.030 | 0.056 | 0.043 | 0.061 | 0.069 | 0.099 | 0.089 | 0.097 | 0.082 | 0.049 | 0.082 | 0.087 | 0.078 | 0.090 | 0.078 |
| 7 | 0.007 | 0.019 | 0.031 | 0.030 | 0.056 | 0.043 | 0.061 | 0.069 | 0.099 | 0.089 | 0.097 | 0.082 | 0.050 | 0.082 | 0.087 | 0.079 | 0.099 | 0.079 |
| 8 | 0.007 | 0.019 | 0.031 | 0.030 | 0.056 | 0.043 | 0.060 | 0.068 | 0.097 | 0.087 | 0.095 | 0.080 | 0.049 | 0.079 | 0.083 | 0.076 | 0.100 | 0.076 |
| 9 | 0.007 | 0.019 | 0.032 | 0.031 | 0.057 | 0.044 | 0.061 | 0.069 | 0.098 | 0.088 | 0.095 | 0.079 | 0.046 | 0.078 | 0.082 | 0.076 | 0.098 | 0.076 |
| Robust Model | | | | | | | | | | | | | | | | | |
| Iteration \ Layer | 1 | 2 | 3 | 4 | 5 | 6 | 7 | 8 | 9 | 10 | 11 | 12 | 13 | 14 | 15 | 16 | 17 | 18 |
| 0 | 0.012 | 0.013 | 0.022 | 0.022 | 0.033 | 0.027 | 0.030 | 0.030 | 0.043 | 0.037 | 0.046 | 0.052 | 0.074 | 0.065 | 0.083 | 0.065 | 0.091 | 0.065 |
| 1 | 0.007 | 0.006 | 0.009 | 0.009 | 0.013 | 0.010 | 0.011 | 0.010 | 0.014 | 0.012 | 0.015 | 0.016 | 0.023 | 0.020 | 0.026 | 0.020 | 0.026 | 0.020 |
| 2 | 0.005 | 0.004 | 0.006 | 0.006 | 0.008 | 0.007 | 0.007 | 0.006 | 0.009 | 0.007 | 0.009 | 0.009 | 0.013 | 0.011 | 0.015 | 0.010 | 0.012 | 0.010 |
| 3 | 0.004 | 0.003 | 0.004 | 0.004 | 0.006 | 0.005 | 0.005 | 0.004 | 0.006 | 0.005 | 0.006 | 0.006 | 0.009 | 0.008 | 0.009 | 0.006 | 0.007 | 0.006 |
| 4 | 0.004 | 0.003 | 0.004 | 0.004 | 0.005 | 0.004 | 0.004 | 0.004 | 0.005 | 0.004 | 0.005 | 0.005 | 0.007 | 0.007 | 0.008 | 0.006 | 0.006 | 0.006 |
| 5 | 0.004 | 0.003 | 0.004 | 0.004 | 0.005 | 0.004 | 0.004 | 0.004 | 0.005 | 0.004 | 0.005 | 0.005 | 0.007 | 0.007 | 0.008 | 0.006 | 0.006 | 0.006 |
| 6 | 0.004 | 0.003 | 0.004 | 0.004 | 0.005 | 0.004 | 0.004 | 0.004 | 0.006 | 0.005 | 0.006 | 0.006 | 0.008 | 0.007 | 0.009 | 0.006 | 0.006 | 0.006 |
| 7 | 0.004 | 0.003 | 0.004 | 0.004 | 0.005 | 0.004 | 0.004 | 0.004 | 0.006 | 0.005 | 0.006 | 0.006 | 0.009 | 0.008 | 0.010 | 0.006 | 0.007 | 0.006 |
| 8 | 0.004 | 0.003 | 0.004 | 0.004 | 0.005 | 0.004 | 0.004 | 0.004 | 0.006 | 0.005 | 0.006 | 0.006 | 0.009 | 0.008 | 0.010 | 0.006 | 0.007 | 0.006 |
| 9 | 0.004 | 0.003 | 0.004 | 0.004 | 0.005 | 0.004 | 0.004 | 0.004 | 0.006 | 0.005 | 0.006 | 0.006 | 0.009 | 0.008 | 0.009 | 0.006 | 0.007 | 0.006 |

Table 1: Activation relative change $\|\boldsymbol{a}_t - \boldsymbol{a}_{t-1}\|/\|\boldsymbol{a}_t\|$ for ResNet-18 on CIFAR-10.

| Normal Model | | | | | | | | | | | | | | | | | |
|---|---|---|---|---|---|---|---|---|---|---|---|---|---|---|---|---|---|---|
| Iteration \ Layer | 1 | 2 | 3 | 4 | 5 | 6 | 7 | 8 | 9 | 10 | 11 | 12 | 13 | 14 | 15 | 16 | 17 | 18 |
| 0 | 0.376 | 0.387 | 0.726 | 0.533 | 0.569 | 0.440 | 0.585 | 0.604 | 0.606 | 0.630 | 0.663 | 0.648 | 0.661 | 0.669 | 0.688 | 0.679 | 0.709 | 0.768 |
| 1 | 0.187 | 0.190 | 0.420 | 0.334 | 0.346 | 0.244 | 0.370 | 0.415 | 0.457 | 0.485 | 0.516 | 0.505 | 0.516 | 0.526 | 0.552 | 0.541 | 0.572 | 0.661 |
| 2 | 0.058 | 0.066 | 0.172 | 0.236 | 0.253 | 0.116 | 0.218 | 0.321 | 0.365 | 0.411 | 0.454 | 0.441 | 0.454 | 0.468 | 0.495 | 0.488 | 0.528 | 0.639 |
| 3 | 0.017 | 0.042 | 0.188 | 0.230 | 0.231 | 0.097 | 0.412 | 0.314 | 0.380 | 0.422 | 0.464 | 0.457 | 0.476 | 0.490 | 0.520 | 0.514 | 0.562 | 0.670 |
| 4 | 0.015 | 0.041 | 0.224 | 0.239 | 0.246 | 0.100 | 0.400 | 0.331 | 0.392 | 0.432 | 0.484 | 0.473 | 0.488 | 0.508 | 0.536 | 0.533 | 0.576 | 0.684 |
| 5 | 0.020 | 0.042 | 0.153 | 0.239 | 0.246 | 0.102 | 0.388 | 0.327 | 0.393 | 0.432 | 0.479 | 0.471 | 0.487 | 0.504 | 0.538 | 0.529 | 0.579 | 0.687 |
| 6 | 0.020 | 0.042 | 0.217 | 0.235 | 0.251 | 0.099 | 0.345 | 0.321 | 0.392 | 0.435 | 0.484 | 0.477 | 0.494 | 0.510 | 0.544 | 0.538 | 0.593 | 0.700 |
| 7 | 0.026 | 0.042 | 0.160 | 0.235 | 0.243 | 0.101 | 0.374 | 0.323 | 0.402 | 0.443 | 0.488 | 0.479 | 0.497 | 0.513 | 0.554 | 0.543 | 0.606 | 0.712 |
| 8 | 0.018 | 0.045 | 0.194 | 0.237 | 0.247 | 0.102 | 0.422 | 0.331 | 0.405 | 0.449 | 0.499 | 0.491 | 0.510 | 0.528 | 0.563 | 0.560 | 0.617 | 0.716 |
| 9 | 0.013 | 0.042 | 0.214 | 0.242 | 0.253 | 0.103 | 0.440 | 0.330 | 0.414 | 0.452 | 0.509 | 0.498 | 0.522 | 0.539 | 0.583 | 0.572 | 0.635 | 0.736 |
| Robust Model | | | | | | | | | | | | | | | | | |
| Iteration \ Layer | 1 | 2 | 3 | 4 | 5 | 6 | 7 | 8 | 9 | 10 | 11 | 12 | 13 | 14 | 15 | 16 | 17 | 18 |
| 0 | 0.058 | 0.183 | 0.402 | 0.231 | 0.195 | 0.181 | 0.245 | 0.218 | 0.233 | 0.232 | 0.245 | 0.232 | 0.231 | 0.227 | 0.235 | 0.233 | 0.235 | 0.270 |
| 1 | 0.011 | 0.086 | 0.067 | 0.102 | 0.096 | 0.097 | 0.169 | 0.136 | 0.154 | 0.157 | 0.159 | 0.155 | 0.153 | 0.154 | 0.171 | 0.159 | 0.163 | 0.201 |
| 2 | 0.007 | 0.058 | 0.040 | 0.069 | 0.098 | 0.093 | 0.155 | 0.125 | 0.143 | 0.148 | 0.152 | 0.145 | 0.145 | 0.145 | 0.149 | 0.148 | 0.153 | 0.189 |
| 3 | 0.006 | 0.050 | 0.054 | 0.071 | 0.096 | 0.094 | 0.163 | 0.130 | 0.142 | 0.142 | 0.156 | 0.144 | 0.147 | 0.144 | 0.147 | 0.147 | 0.150 | 0.189 |
| 4 | 0.006 | 0.064 | 0.057 | 0.084 | 0.106 | 0.092 | 0.166 | 0.128 | 0.140 | 0.141 | 0.150 | 0.141 | 0.143 | 0.144 | 0.145 | 0.147 | 0.150 | 0.192 |
| 5 | 0.005 | 0.071 | 0.054 | 0.093 | 0.100 | 0.094 | 0.160 | 0.128 | 0.139 | 0.141 | 0.148 | 0.141 | 0.140 | 0.140 | 0.147 | 0.143 | 0.151 | 0.188 |
| 6 | 0.005 | 0.068 | 0.142 | 0.087 | 0.095 | 0.093 | 0.173 | 0.133 | 0.155 | 0.155 | 0.162 | 0.157 | 0.154 | 0.159 | 0.154 | 0.159 | 0.160 | 0.198 |
| 7 | 0.007 | 0.065 | 0.122 | 0.080 | 0.103 | 0.095 | 0.174 | 0.138 | 0.148 | 0.155 | 0.166 | 0.157 | 0.157 | 0.159 | 0.162 | 0.161 | 0.163 | 0.200 |
| 8 | 0.006 | 0.065 | 0.034 | 0.077 | 0.106 | 0.097 | 0.187 | 0.143 | 0.158 | 0.162 | 0.166 | 0.162 | 0.161 | 0.165 | 0.162 | 0.167 | 0.168 | 0.204 |
| 9 | 0.005 | 0.059 | 0.052 | 0.081 | 0.099 | 0.095 | 0.173 | 0.137 | 0.151 | 0.158 | 0.165 | 0.159 | 0.156 | 0.160 | 0.161 | 0.163 | 0.164 | 0.201 |

Table 2: Gradient relative change $\|\nabla_{\boldsymbol{o}_t}\mathcal{L} - \nabla_{\boldsymbol{o}_{t-1}}\mathcal{L}\|/\|\nabla_{\boldsymbol{o}_t}\mathcal{L}\|$ for ResNet-18 on CIFAR-10.

# B  ADVERSARIAL TRAINING

We integrate Spiking-PGD into adversarial training to reduce the cost of generating perturbations. In this setting, we investigate two scheduling strategies for controlling the spiking threshold $\rho$: a constant threshold and an exponential decay schedule. For the latter, the threshold at epoch $t$ is defined as $\rho(t) = \rho_0 \cdot \frac{e^{-\lambda t/N} - e^{-\lambda}}{1 - e^{-\lambda}}$, where $\rho_0 = 0.1$ is the initial threshold, $N = 200$ is the total number of epochs, and $\lambda$ is a hyperparameter that controls the decay rate. By construction, $\rho(0) = \rho_0$ and $\rho(T) = 0$.

**Performance Analysis.**  Table 3 summarizes clean accuracy, robust accuracy, their sum, and the effective precision under different schedules (constant schedule and exponential decay schedule). Several trends emerge:

- **Efficiency:** Spike-PGD achieves substantial computational savings while maintaining accuracy comparable to standard PGD-based adversarial training.

- **Stability:** Exponential decay schedules consistently outperform constant thresholds, offering smoother convergence and better robustness.

- **Trade-off control:** By tuning $\lambda$, practitioners can balance computational cost against training precision. Larger $\lambda$ yields faster threshold decay, higher precision, and more accurate training dynamics, at the cost of increased computation.

| Method | | Clean (%) | Robust (%) | Sum. (%) | Precision (%) |
|---|---|---|---|---|---|
| Vanilla PGD-AT | | 80.18 | 48.75 | 128.93 | 100 |
| Constant | $\rho = 0.0$ | 80.18 | 48.75 | 128.93 | 100 |
| | $\rho = 0.01$ | 80.47 | 49.04 | 129.51 | 98.9934369883219 |
| | $\rho = 0.015$ | 79.64 | 48.52 | 128.16 | 95.71530061299882 |
| | $\rho = 0.02$ | 80.0 | 48.63 | 128.63 | 64.68025277743948 |
| | $\rho = 0.025$ | 80.52 | 48.78 | 129.3 | 45.20519898780769 |
| | $\rho = 0.03$ | 81.38 | 45.37 | 126.75 | 32.11615854071097 |
| | $\rho = 0.035$ | 82.11 | 41.15 | 123.26 | 51.55962868239082 |
| | $\rho = 0.04$ | 85.47 | 39.95 | 125.42 | 52.96381547788197 |
| | $\rho = 0.045$ | 88.38 | 39.72 | 128.1 | 26.003423591 65889 |
| | $\rho = 0.05$ | 86.48 | 37.59 | 124.07 | 14.28571428571428 |
| Exponential | $\lambda = 1.0$ | 85.25 | 43.66 | 128.91 | 28.44629832609373 |
| | $\lambda = 2.0$ | 82.33 | 47.45 | 129.78 | 44.056306580603255 |
| | $\lambda = 3.0$ | 81.09 | 48.88 | **129.97** | 55.72839957238934 |
| | $\lambda = 4.0$ | 80.75 | 48.49 | 129.24 | 64.77576151233441 |
| | $\lambda = 5.0$ | 80.46 | 48.76 | 129.22 | 71.58324199244916 |
| | $\lambda = 6.0$ | 80.0 | 48.64 | 128.64 | 71.76493592606124 |
| | $\lambda = 7.0$ | 80.4 | 48.87 | 129.27 | 77.41146700225985 |
| | $\lambda = 8.0$ | 80.27 | 48.69 | 128.96 | 78.97461400018945 |
| | $\lambda = 9.0$ | 80.19 | 48.59 | 128.78 | 81.21360234915223 |
| | $\lambda = 10.0$ | 80.12 | 48.77 | 128.89 | 83.07328921906928 |

Table 3: Adversarial training results on CIFAR-10. Spike-PGD with exponential decay schedules consistently achieves competitive performance with significantly reduced computation.

**Accuracy Curves.** To better understand the training dynamics under different schedules, we visualize the evolution of accuracy across epochs during adversarial training with Spiking-PGD in Figure 10 and Figure 11. Both clean and robust accuracy are tracked throughout training. The results show that the exponential decay schedule leads to more consistent final performance with baseline compared to the constant schedule.

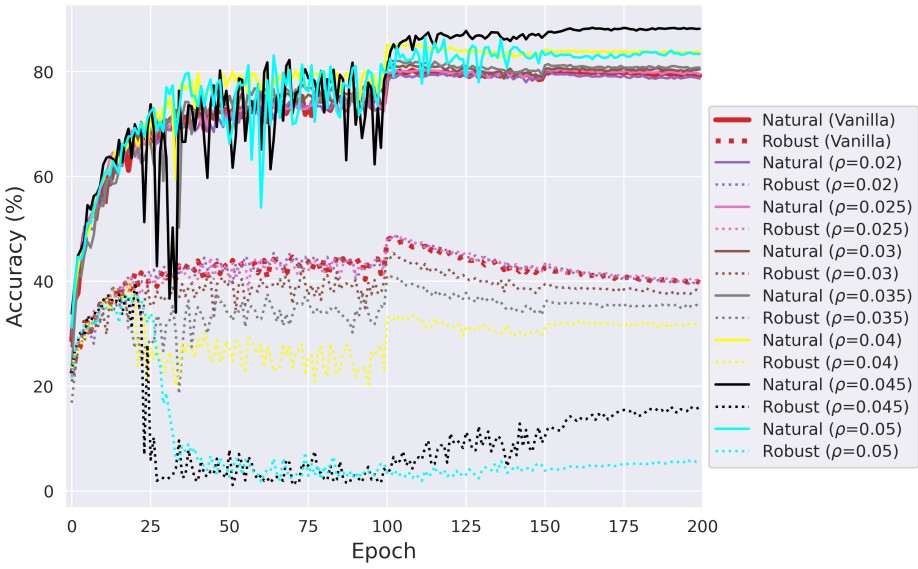

Figure 10: Accuracy curves of adversarial training on CIFAR-10. Clean (solid) and robust (dotted) accuracy for vanilla PGD-AT and Spiking-PGD with different threshold $\rho$ in constant schedule.

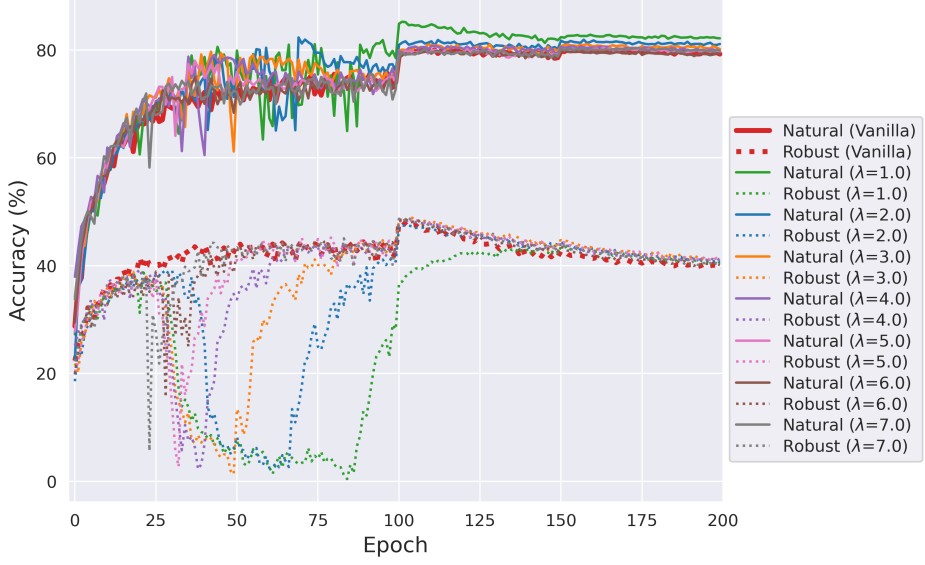

Figure 11: Accuracy curves of adversarial training on CIFAR-10. Clean (solid) and robust (dotted) accuracy for vanilla PGD-AT and Spiking-PGD with different decay rate $\lambda$ in exponential decay schedule.

**Precision Curves.** Furthermore, we visualize the evolution of precision across epochs during adversarial training with Spiking-PGD in Figure 12 and Figure 13, corresponding to the constant and exponential decay schedules, respectively. The results show that the exponential decay schedule achieves more stable precision control throughout training, whereas the constant schedule is highly sensitive to the threshold setting.

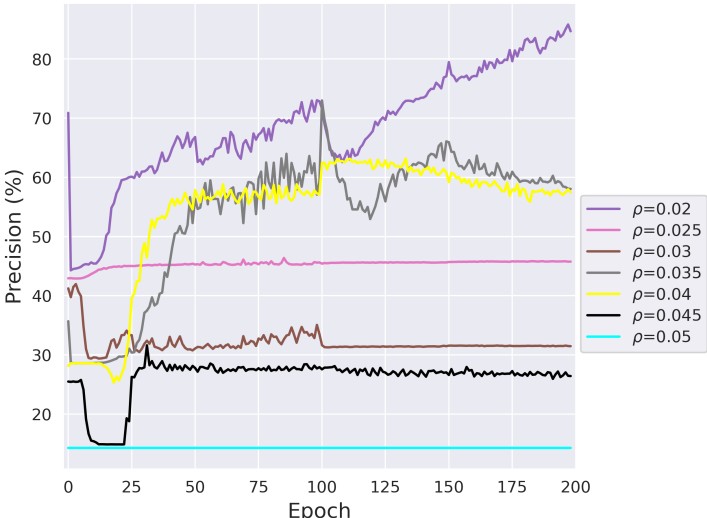

Figure 12: Precision curves of adversarial training on CIFAR-10 with different threshold $\rho$ in constant schedule.

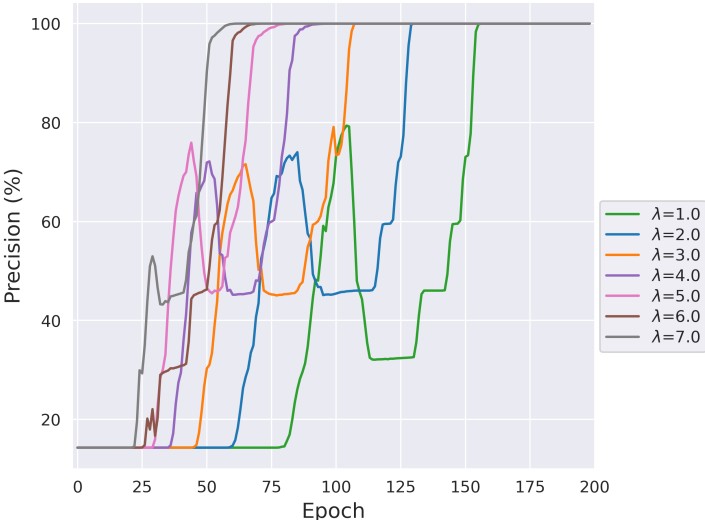

Figure 13: Precision curves of adversarial training on CIFAR-10 with different decay rate $\lambda$ in exponential decay schedule.

## C  THEORETICAL PROOF

### C.1  PROOF OF PROPOSITION 4.1

**Proposition 4.1.** Let $V_{\mathrm{coarse}}$ and $V_{\mathrm{fine}}$ denote the optimal objective values of Eq. (1) and Eq. (2), respectively, then we have:
$$V_{\mathrm{coarse}} \leq V_{\mathrm{fine}}.$$
The feasible set of Eq. (1) can be embedded into the feasible set of Eq. (2) by masks that compute layers for iterations $t \leq S$ and skip layers for $t > S$.

*Proof.* For any $S \in \{0, \dots, T\}$ feasible in Eq. (1) (i.e., $\sum_{t=1}^{S} C_t \leq C_{\mathrm{total}}$), define the corresponding mask:
$$\delta_{t,l}^{(S)} = \begin{cases} 1, & t \leq S, \\ 0, & t > S, \end{cases} \qquad \forall\, t \in \{1, \dots, T\},\ l \in \{1, \dots, L\}.$$

The cost of $\Delta^{(S)}$ equals $\sum_{t=1}^{S} \sum_{l=1}^{L} C_{t,l} = \sum_{t=1}^{S} C_t$, hence $\Delta^{(S)}$ is feasible for Eq. (2). Moreover, executing the attack with $\Delta^{(S)}$ results in no updates after iteration $S$, so $\boldsymbol{x}_{T+1}(\Delta^{(S)}) = \boldsymbol{x}_{S+1}$. Thus every feasible $S$ for Eq. (1) admits a corresponding feasible mask $\Delta^{(S)}$ with identical objective value. Therefore the optimum over all $S$ cannot exceed the optimum over all masks $\Delta$, i.e. $V_{\mathrm{coarse}} \leq V_{\mathrm{fine}}$. $\qquad\square$

