# OpenReview forum: "Fine-Grained Iterative Adversarial Attacks with Limited Computation Budget"
_ICLR.cc/2026/Conference — ICLR 2026 Poster_

### Official Review · Reviewer_3r6R · 2025-10-20

**Soundness:** 3
**Presentation:** 4
**Contribution:** 3
**Rating:** 8
**Confidence:** 4

**Summary:**

This paper presents a timely approach to improving the computational efficiency of iterative adversarial attacks. To this end, authors replace the coarse-grained control of iteration count with a fine-grained, layer-wise spiking mechanism.

**Strengths:**

The topic this paper focused on is very interesting and novel. Besides, this paper is well-written.

**Weaknesses:**

1. Authors should report the magnitude of the error brought by using S_{\rho}. That is, authors should examine the approximation error between the left and right part of Figure 3.

2. Can the proposed method applied to the attack on large-scale dataset, such as the ImageNet dataset?

3. Can the proposed method be well generalized to more iterative attacks, such as C&W? It would be better authors add a discussion for the application to the black-box attacks.

minor:
1. The condition in Algorithm 2 "if $t=1$ or ..." seems to have a typo (likely should be if $t>1$ or ... for the second case). The notation $∂^(l)_t$ is also not explicitly defined before its use. A more polished and commented algorithm would improve clarity.

**Questions:**

Please refer to the weaknesses.

---

> ### Author Response · Authors · 2025-11-24
>
> **W1: Authors should report the magnitude of the error brought by using $S_\rho$. That is, authors should examine the approximation error between the left and right part of Figure 3.**
>
> **A:** We thank the reviewer for pointing this out. The approximation error between the left and right side of Figure 3 corresponds to the residual term omitted when  $S_\rho(a_t, a_{t-1}) = 0$,  i.e., when the update is skipped.
>
> Our spiking update reuses $o_{t-1}^{(l)}$ whenever  $\|\|a_t^{(l)} - a_{t-1}^{(l)}\|\| / \|\|a_t^{(l)}\|\| < \rho$,  so when the update is skipped the approximation error is $e_t^{(l)} = A^{(l)}(a_t^{(l)} - a_{t-1}^{(l)})$.
>
> Under this trigger condition, the magnitude of this error is naturally bounded as
> $\|\|e_t^{(l)}\|\| = \bigl\|\bigl\| A^{(l)}(a_t^{(l)} - a_{t-1}^{(l)}) \bigr\|\bigr\|
> \le \|\|A^{(l)}\|\| \cdot \|\|a_t^{(l)} - a_{t-1}^{(l)}\|\|
> < \rho \|\|A^{(l)}\|\| \cdot \|\|a_t^{(l)}\|\|,$
>
> showing that the error introduced by $S_\rho$ is explicitly upper-bounded by the threshold $\rho$ and the operator norm of the layer.
>
> **W2: Can the proposed method be applied to the attack on large-scale dataset, such as the ImageNet dataset?**
>
> **A:** Thank you for the thoughtful comment. Our approach can be applied to large-scale datasets like ImageNet, and even datasets beyond the vision domain. To substantiate this claim, we additionally evaluate Spiking-PGD on Vision Transformers (ViT) on ImageNet and Language Transformers (BERT) on SST-2. Across both cases, Spiking-PGD consistently achieves significantly higher attack strength under comparable computation cost, demonstrating its strong generalization.
>
> ---
>
> ### (1) Vision Transformers (ViT) on ImageNet
>
> |                |  |  | | |  | |  | |  |  |  |
> |----------------|----|----|----|----|----|----|----|----|----|----|-----|
> | **PGD**        |    |    |    |    |    |    |    |    |    |    |     |
> | Computation Cost (%)  |    | 10 | 20 | 30 | 40 | 50 | 60 | 70 | 80 | 90 | 100 |
> | Accuracy (%)   |    | 48.91 | 35.72 | 29.91 | 25.08 | 23.97 | 21.55 | 20.03 | 19.57 | 19.34 | 18.55 |
> | **Spiking-PGD**|    |    |    |    |    |    |    |    |    |    |     |
> | Computation Cost  (%)  |    | 6.06 | 13.35 | 32.70 | 55.02 | 60.06 | 67.72 | 81.93 | 87.23 | 90.48 |
> | Accuracy (%)   |    | 41.65 | 32.56 | 24.10 | 21.84 | 20.03 | 19.41 | 18.72 | 18.55 | 18.55 |
>
> **Table 1:** Spiking-PGD on ViT (ImageNet).
>
> ---
>
> ### (2) Language Transformers (BERT) on SST-2
>
> |                |  |  | | |  | |  | |  |  |  |
> |----------------|----|----|----|----|----|----|----|----|----|----|-----|
> | **PGD**        |    |    |    |    |    |    |    |    |    |    |     |
> | Computation Cost  (%)  |    | 10 | 20 | 30 | 40 | 50 | 60 | 70 | 80 | 90 | 100 |
> | Accuracy (%)   |    | 89.9 | 89.9 | 86.87 | 83.84 | 78.79 | 75.76 | 72.73 | 66.67 | 64.65 | 57.58 |
> | **Spiking-PGD**|    |    |    |    |    |    |    |    |    |    |     |
> | Computation Cost  (%)  |    | 5.21 | 9.42 | 11.90 | 30.80 | 49.51 | 65.96 | 80.65 | 88.64 | 90.48 | — |
> | Accuracy (%)   |    | 88.89 | 86.87 | 82.83 | 80.81 | 72.73 | 69.70 | 61.62 | 58.59 | 57.58 | — |
>
> **Table 2:** Spiking-PGD on BERT (SST-2).
>
> ---

---

> ### Author Response · Authors · 2025-11-24
>
> **W3: Can the proposed method be well generalized to more iterative attacks, such as C&W? It would be better if authors add a discussion for the application to black-box attacks.**
>
> **A:** Thanks for your comments. Our Spiking-Attack is a general framework that can be plugged into a wide range of iterative attacks, including C&W and black-box attacks.
>
> ---
>
> ### **Application to C&W attack**
> The C&W attack also generates a sequence of iterates {$x_t$} through iterative optimization, and the temporal activation patterns across successive iterates remain highly correlated. Thus, the spiking mechanism can skip redundant forward/backward computations whenever the activation change across iterations is small. This results in substantial computational savings without affecting convergence, as demonstrated in the table below.
>
> #### **C&W**
> | Computation Cost (%) | 10 | 20 | 30 | 40 | 50 | 60 | 70 | 80 | 90 | 100 |
> |---------------|----|----|----|----|----|----|----|----|----|-----|
> | Accuracy (%)  | 74.91 | 65.57 | 56.12 | 48.08 | 40.81 | 34.37 | 28.42 | 23.56 | 19.50 | 15.98 |
>
> #### **Spiking-C&W**
> | Computation Cost (%) | 5.31 | 16.12 | 21.21 | 38.23 | 48.11 | 59.30 | 75.52 | 85.59 | 95.19 | 100.00 |
> |---------------|------|-------|-------|-------|-------|-------|-------|-------|--------|--------|
> | Accuracy (%)  | 69.84 | 59.83 | 46.41 | 39.56 | 32.26 | 21.62 | 19.05 | 17.75 | 16.55 | 15.98 |
>
> **Table 17: Spiking-C&W (ViT, ImageNet, $\epsilon = 2/255$).**
>
> ---
>
> ### **Discussion on black-box attacks**
> Our method is directly applicable to black-box attacks that rely on iterative gradient estimation, such as NES or bandit-based attacks, because these methods also generate a sequence of inputs $\{x_t\}$ that exhibit strong temporal correlation. The spiking mechanism can therefore reduce the number of expensive forward evaluations of the victim model. We will add a short discussion in the revision and comprehensive empirical evaluation of spiking-based black-box attacks is an exciting future direction.
>
>
> **W4: The condition in Algorithm 2 “if $t = 1$ or …” seems to have a typo (likely should be “if $t > 1$ or …” for the second case). The notation $\partial_t^{(l)}$ is also not explicitly defined before its use. A more polished and commented algorithm would improve clarity.**
>
> **A:** Thanks for pointing out this minor typo. We will correct it in the revision. Regarding the notation $\partial$, it denotes the standard derivative. For example,  $\frac{\partial \mathcal{L}}{\partial a_t^{(l)}}$
> represents the derivative of the loss $\mathcal{L}$ with respect to the activation $a$ at iteration $t$ and layer $l$.

---

> ### Comment · Reviewer_3r6R · 2025-11-27
> **Response**
>
> Thanks for authors' response. I will keep my rating unchanged.
> By the way, i think it would be better could report the empirical approximation error in practice, rather than a theoretical bound.

---

> > ### Author Response · Authors · 2025-11-30
> >
> > Thank you for your valuable suggestion. We believe that reporting the empirical approximation error is important and complements the theoretical analysis. Recall that the theoretical bound gives
> >
> > $
> > \|\|e_t^{(l)}\|\| = \|\|\mathcal{A}^{(l)}(a_t^{(l)} - a_{t-1}^{(l)})\|\|
> > \le
> > \|\|\mathcal{A}^{(l)}\|\| \cdot \|\|a_t^{(l)} - a_{t-1}^{(l)}\|\|
> > <
> > \rho  \|\|\mathcal{A}^{(l)}\| \cdot \|a_t^{(l)}\|\|.
> > $
> >
> > To validate this bound in practice, we further quantify the empirical approximation error under different values of $\rho$. The results are summarized below:
> >
> > | $\rho (\times 10^{-3})$ | $\|e_t^{(l)}\|$ |
> > |-------------------------|-----------------|
> > | 6  | 4.5933 |
> > | 7  | 5.6246 |
> > | 8  | 6.3197 |
> > | 9  | 7.1141 |
> > | 10 | 7.8521 |
> > | 11 | 8.5468 |
> > | 12 | 9.2277 |
> > | 13 | 9.9325 |
> > | 14 | 10.6299 |
> > | 15 | 11.7119 |
> >
> > **Table:** Empirical approximation error $\|\|e_t^{(l)}\|\|$ under different values of $\rho$.
> >
> > As shown in the table, the empirical approximation error grows monotonically with $\rho$, which is consistent with the theoretical bound and confirms that our approximation behaves as predicted in practice.

---

### Official Review · Reviewer_uH8u · 2025-10-27

**Soundness:** 4
**Presentation:** 3
**Contribution:** 3
**Rating:** 8
**Confidence:** 4

**Summary:**

This paper tackles the efficiency problem of iterative adversarial attacks, which are computationally expensive due to repeated forward and backward passes. The authors propose Spiking-PGD, featuring Spiking Forward Computation, which skips redundant layer computations based on activation changes, and Virtual Surrogate Gradient, which restores gradient flow when activations are reused. Formulated as a layer–iteration joint optimization problem, this approach enables fine-grained computation allocation. Experiments on multiple datasets show that Spiking-PGD significantly improves attack success rates under the same computational budget and reduces adversarial training costs by up to 70% without sacrificing performance.

**Strengths:**

1.The work demonstrates strong originality and innovation.
2.The method is elegantly designed.
3.The paper is well-structured and clearly written.

**Weaknesses:**

1.The paper lacks a released code repository, which limits reproducibility.
2.My main concern is that since the proposed method reuses and stores previous activations, it is likely to increase memory consumption. As GPU memory is often a more critical bottleneck than time, corresponding experiments or analyses on memory overhead should be included.
3.The vision experiments are limited to ResNet-18; it is recommended to include and discuss more diverse architectures, especially Transformer-based models.
4.The evaluation should be extended to more diverse modalities, such as audio and NLP datasets.

**Questions:**

1. Please compare the GPU memory overhead with baselines.

---

> ### Author Response · Authors · 2025-11-24
>
> **W1: The paper lacks a released code repository, which limits reproducibility.**
>
> **A:** Thanks for pointing out. We have put our anonymous code repository here:
> https://anonymous.4open.science/r/spiking_attack-E5F3.
>
> ---
>
> **W2: My main concern is that since the proposed method reuses and stores previous activations, it is likely to increase memory consumption. As GPU memory is often a more critical bottleneck than time, corresponding experiments or analyses on memory overhead should be included.**
>
> **A:** We thank the reviewer for raising the concern about potential GPU memory overhead. Importantly, the proposed method stores only the activations from **one previous iteration** (two small buffers per layer). These buffers are overwritten at each step, so the memory cost does **not grow with the number of attack iterations $T$.** Thus, both the forward and backward passes still keep only a single computation graph, and the dominant memory usage remains the same as standard PGD.
>
> To verify this empirically, we explicitly measured GPU memory usage using standard PyTorch APIs:
> - `torch.cuda.memory_allocated()` (current usage)
> - `torch.cuda.max_memory_allocated()` (peak usage during the attack)
>
> The peak value reflects the actual worst-case footprint of the forward–backward computation, which determines whether the attack fits into GPU memory.
>
> The following table reports the measured **peak GPU memory** for PGD and Spiking-PGD under different numbers of attack iterations $T$. The results show:
>
> | Method               | Peak GPU mem |
> |----------------------|--------------|
> | PGD                  | 1968.12 MB   |
> | Spiking PGD (T = 5)  | 2101.73 MB   |
> | Spiking PGD (T = 10) | 2101.73 MB   |
> | Spiking PGD (T = 15) | 2101.73 MB   |
> | Spiking PGD (T = 20) | 2101.73 MB   |
>
>
> The increase is a small, constant overhead (≈ 133 MB, or ≈ 7%) and does **not** scale with the number of attack iterations. This confirms that our method introduces only a minor fixed memory cost while providing substantial computational savings across many attack steps.

---

> ### Author Response · Authors · 2025-11-24
>
> **W3 & W4: The vision experiments are limited to ResNet-18; it is recommended to include and dis-
> cuss more diverse architectures, especially Transformer-based models. The evaluation should be extended to more diverse modalities, such as audio and NLP datasets**
>
>
> **A:** Thank you for the thoughtful comment. Our approach is highly architecture-agnostic and dataset-agnostic. To substantiate this claim, we additionally evaluate Spiking-PGD on four substantially different model families and tasks:
> (i) Vision Transformers (ViT) on ImageNet;
> (ii) Language Transformers (BERT) on SST-2;
> (iii) VGG on CIFAR-10;
> (iv) Large-scale CNNs (ResNet-50) on CIFAR-10.
>
> Across all cases, Spiking-PGD consistently achieves significantly better strength under comparable computation cost, demonstrating its strong generalization.
>
> ---
>
> ### (1) Vision Transformers (ViT) on ImageNet
>
> |                |  |  | | |  | |  | |  |  |  |
> |----------------|----|----|----|----|----|----|----|----|----|----|-----|
> | **PGD**        |    |    |    |    |    |    |    |    |    |    |     |
> | Computation Cost (%)  |    | 10 | 20 | 30 | 40 | 50 | 60 | 70 | 80 | 90 | 100 |
> | Accuracy (%)   |    | 48.91 | 35.72 | 29.91 | 25.08 | 23.97 | 21.55 | 20.03 | 19.57 | 19.34 | 18.55 |
> | **Spiking-PGD**|    |    |    |    |    |    |    |    |    |    |     |
> | Computation Cost (%)  |    | 6.06 | 13.35 | 32.70 | 55.02 | 60.06 | 67.72 | 81.93 | 87.23 | 90.48 |
> | Accuracy (%)   |    | 41.65 | 32.56 | 24.10 | 21.84 | 20.03 | 19.41 | 18.72 | 18.55 | 18.55 |
>
> **Table 1:** Spiking-PGD on ViT (ImageNet).
>
> ---
>
> ### (2) Language Transformers (BERT) on SST-2
>
> |                |  |  | | |  | |  | |  |  |  |
> |----------------|----|----|----|----|----|----|----|----|----|----|-----|
> | **PGD**        |    |    |    |    |    |    |    |    |    |    |     |
> | Computation Cost (%)  |    | 10 | 20 | 30 | 40 | 50 | 60 | 70 | 80 | 90 | 100 |
> | Accuracy (%)   |    | 89.9 | 89.9 | 86.87 | 83.84 | 78.79 | 75.76 | 72.73 | 66.67 | 64.65 | 57.58 |
> | **Spiking-PGD**|    |    |    |    |    |    |    |    |    |    |     |
> | Computation Cost (%)  |    | 5.21 | 9.42 | 11.90 | 30.80 | 49.51 | 65.96 | 80.65 | 88.64 | 90.48 | — |
> | Accuracy (%)   |    | 88.89 | 86.87 | 82.83 | 80.81 | 72.73 | 69.70 | 61.62 | 58.59 | 57.58 | — |
>
> **Table 2:** Spiking-PGD on BERT (SST-2).
>
> ---
>
> ### (3) VGG on CIFAR-10
> |                |  |  | | |  | |  | |  |  |  |
> |----------------|----|----|----|----|----|----|----|----|----|----|-----|
> | **PGD**        |    |    |    |    |    |    |    |    |    |    |     |
> | Computation Cost (%) |    | 10 | 20 | 30 | 40 | 50 | 60 | 70 | 80 | 90 | 100 |
> |   Accuracy (%)      |    | 57.74 | 46.96 | 43.99 | 42.74 | 39.10 | 37.01 | 35.51 | 34.48 | 32.41 | 31.23 |
> | **Spiking-PGD**|    |    |    |    |    |    |    |    |    |    |     |
> | Computation Cost (%) |    | 18.55 | 26.75 | 32.35 | 41.50 | 59.70 | 72.40 | 84.35 | 96.15 | 98.60 | 100.00 |
> | Accuracy (%)   |    | 40.86 | 39.48 | 38.81 | 38.01 | 35.68 | 34.46 | 33.08 | 31.80 | 31.44 | 31.23 |
>
> **Table 3:** Spiking-PGD on VGG11 (CIFAR-10).
>
> ---
>
> ### (4) Large ResNets (ResNet-50) on CIFAR-10
>
> |                |   |    |    |   |     |    |    |    |    |   |   |    |
> |----------------|--------|--------|--------|--------|--------|--------|--------|--------|--------|--------|--------|--------|
> | **PGD**        |        |        |        |        |        |        |        |        |        |        |        |        |
> | Computation Cost (%) | 10    | 20    | 30    | 40    | 50    | 60    | 70    | 80    | 90    | 100 |      |      |
> | Accuracy (%)   | 87.83 | 82.25 | 74.89 | 69.16 | 64.94 | 62.14 | 59.95 | 57.93 | 57.11 | 55.83 |      |      |
> | **Spiking-PGD**|        |        |        |        |        |        |        |        |        |        |        |        |
> |Computation Cost (%) | 5.00  | 10.10 | 11.95 | 18.55 | 30.30 | 44.19 | 56.45 | 64.78 | 69.90 | 72.83 | 85.34 | 100.00 |
> | Accuracy (%)   | 69.71 | 68.71 | 67.95 | 66.76 | 65.50 | 65.02 | 65.12 | 63.60 | 59.23 | 56.62 | 55.97 | 55.83 |
>
> **Table 4:** Spiking-PGD on ResNet-50 (CIFAR-10).
>
> ---
>
> Overall, these results confirm that our method reliably generalizes to diverse model architectures—including both CNNs and Transformers—and across both vision and language tasks.
>
> **Q1: Please compare the GPU memory overhead with baselines.**
>
> **A:** See W2.

---

> > ### Comment · Reviewer_uH8u · 2025-11-25
> >
> > I already read the response, and keep the rating as 8. One additional suggestion is that maybe change the accuracy with attack success rate, it is more intuitive for attack evaluation.

---

> > > ### Author Response · Authors · 2025-11-25
> > >
> > > We sincerely appreciate the reviewer’s recognition of our work and the valuable suggestions provided. We will take these suggestions into full consideration in our revision.

---

### Official Review · Reviewer_jTdB · 2025-10-27

**Soundness:** 3
**Presentation:** 3
**Contribution:** 3
**Rating:** 4
**Confidence:** 4

**Summary:**

The paper proposes Spiking Iterative Attacks, a method to make iterative adversarial attacks more efficient under limited computation. By observing that layer activations have minor changes across attack iterations, the authors introduce fine-grained, iteration and layer-wise control that recomputes activations only when necessary, and use a virtual surrogate gradient to maintain gradient flow when activations are reused. This approach achieves comparable or stronger attacks at significantly lower computational cost and also speeds up adversarial training.

**Strengths:**

- The paper is well-written and easy to read.
- The contribution is novel and interesting.

**Weaknesses:**

**Missing important baselines induces vague lack of contemporaneity:** The paper compares against standard iterative attacks (PGD, MI-FGSM, I-FGSM) but omits stronger, widely-used robust evaluation suites such as AutoAttack (AA) [ext_ref_1]. Without AA (or similar), it’s hard to evaluate the actual impact of Spiking-PGD. AutoAttack is now a de-facto standard for robust evaluation, and since it only allows modifying iterations, skipping it weakens the claim that Spiking-PGD genuinely expands the efficiency–effectiveness frontier. Unfortunately, the same can be said for the choice of using only ResNet18 models and usual datasets and not extending to transformers. Even by accepting the choice of not using modern architectures or datasets, to broaden the impact of the proposed approach it is significant to extend beyond the single ResNet18 architecture with different Deep Neural Network architectures, such as VGG and others.

**Slight lack of clarity in some aspects:**
- They show activations quickly become similar across iterations, motivating their methodology. Yet, iterative attacks still improve with more iterations. The paper lacks a clear mechanistic explanation reconciling these two facts. If activations stabilize, why do further iterations still yield stronger adversaries?
- Equation (2) is a discrete, combinatorial, non-convex optimization (NP-hard). The paper frames this but then presents a limited discussion on why other well-known strategies (continuous relaxations, differentiable masks, Gumbel-Softmax, integer programming relaxations, STE, or pruning literature methods) are unsuitable, or how they compare.

[ext_ref_1]: Croce, Francesco, and Matthias Hein. "Reliable evaluation of adversarial robustness with an ensemble of diverse parameter-free attacks." International conference on machine learning. PMLR, 2020.

**Questions:**

1. Why did the authors choose not to include AutoAttack (AA) [ext_ref_1] or other modern robust evaluation suites in their experiments, given that these are widely considered standard baselines for assessing adversarial robustness today? Can the authors integrate AA into their experiments during rebuttal?
2. Can the authors test the method on other architectures (e.g., VGG, WideResNet, or even better on transformer-based models) to validate whether the proposed approach generalizes across architectures?
3. The authors show that activations across iterations become highly similar, yet iterative attacks continue to improve as iterations increase. How do the authors reconcile this apparent contradiction? What mechanisms explain the continued effectiveness of attacks even when activations stabilize?
4. Can the authors elaborate more on why do standard relaxations or methods fail in solving equation 2?

Overall, I liked the way in which the paper was presented and the provided method/contribution. However, there are some important limitations from the experimental perspective, as well as concepts that need further elaboration from my point of view, thus justifying my score. I am open to increase my score, however, based on how the authors will conduct rebuttal.

---

> ### Author Response · Authors · 2025-11-24
>
> **W1: The paper compares against standard iterative attacks (PGD, MI-FGSM, I-FGSM) but omits stronger, widely-used robust evaluation suites such as AutoAttack (AA) [1]. Without AA (or similar), it is hard to evaluate the actual impact of Spiking-PGD. AutoAttack is now a de-facto standard for robust evaluation, and since it only allows modifying iterations, skipping it weakens the claim that Spiking-PGD genuinely expands the efficiency–effectiveness frontier.**
>
> [1] Croce, Francesco, and Matthias Hein. *Reliable evaluation of adversarial robustness with an ensemble of diverse parameter-free attacks.* ICML, 2020.
>
> ---
>
> **A:** Thank you for the insightful comments. Let me address all concerns as follows:
>
>
>
> - **AA is significantly more expensive than standard iterative attacks**
>
> AutoAttack (AA) is indeed a strong and widely-used robustness evaluation suite.
> However, it is computationally expensive because it ensembles four heterogeneous attacks: `apgd-ce` (default: n_iter = 100) , `apgd-t` (default: n_iter = 100), `fab-t` (default: n_iter = 100), `square` (default: n_queries = 5000). In contrast, PGD in our paper only need 20 iteration.
>
>
>
> - **Spiking-PGD is a lightweight acceleration framework that can be incorporated into AA**
>
> Spiking-PGD is **not** intended to replace AA.  Instead, it accelerates iterative attacks by reusing redundant gradient updates.
> Thus, Spiking-PGD is **complementary** to AA:  it can be plugged into AA’s iterative components to improve efficiency while maintaining attack effectiveness.
>
> - **AA is unsuitable for adversarial training due to cost**
>
> AA is valuable for **evaluation**, not for **training**, because generating adversarial examples with AA is prohibitively expensive.
> This aligns with the literature: AA is almost exclusively used for testing robustness.
>
>  - **Empirical evaluation of AutoAttack**
>
> To validate our statement, we compare the empirical results on AutoAttack, PGD, and our Spiking-PGD under matched compute. To make a fair comparison, we define the precision as  $T/T_0 \times 100$%, $\quad T_0 = 20$.
> For the ensemble attack AutoAttack, the attacks are executed sequentially following  `apgd-ce → apgd-t → fab-t → square`.  So we allocate $T$ evenly across the four sub-attacks. The results below show that under equal cost, Spiking-PGD achieves stronger attack effectiveness than both AA and vanilla PGD.
>
> ---
>
> #### **AutoAttack**
> | Computation Cost (%) | 20 | 40 | 60 | 80 | 100 |
> |---------------|----|----|----|----|-----|
> | Accuracy (%)  | 48.14 | 46.64 | 44.83 | 43.43 | 42.23 |
>
> #### **PGD**
> | Computation Cost (%) | 10 | 20 | 30 | 40 | 50 | 60 | 70 | 80 | 90 | 100 |
> |---------------|----|----|----|----|----|----|----|----|----|-----|
> | Accuracy (%)  | 63.88 | 52.13 | 44.8 | 41.56 | 40.44 | 39.84 | 39.46 | 39.26 | 39.14 | 38.87 |
>
> #### **Spiking-PGD**
> | Computation Cost (%) | 5.00 | 5.15 | 9.00 | 13.30 | 19.80 | 26.25 | 32.75 | 38.22 | 42.19 | 51.84 | 74.97 |
> |---------------|------|------|------|-------|--------|--------|--------|--------|--------|--------|--------|
> | Accuracy (%)  | 49.12 | 49.03 | 47.21 | 45.41 | 43.65 | 42.52 | 41.93 | 41.78 | 41.53 | 40.43 | 39.31 |
>
> **Table 5: AutoAttack (ResNet18, CIFAR10, budget = 8/255)**
>
> ---
>
> In summary, AA is too expensive to serve as a primary comparative baseline for iterative attacks or adversarial training, but our additional AA-based evaluation confirms that Spiking-PGD improves the efficiency–effectiveness tradeoff even under the AA setting.

---

> ### Author Response · Authors · 2025-11-24
>
> **W2: Unfortunately, the same can be said for the choice of using only ResNet18 models and
> usual datasets and not extending to transformers. Even by accepting the choice of not using
> modern architectures or datasets, to broaden the impact of the proposed approach it is signif-
> icant to extend beyond the single ResNet18 architecture with different Deep Neural Network
> architectures, such as VGG and others.**
>
>
> **A:** Thank you for the thoughtful comment. Our approach is highly architecture-agnostic and dataset-agnostic. To substantiate this claim, we additionally evaluate Spiking-PGD on four substantially different model families and tasks:
> (i) Vision Transformers (ViT) on ImageNet;
> (ii) Language Transformers (BERT) on SST-2;
> (iii) VGG on CIFAR-10;
> (iv) Large-scale CNNs (ResNet-50) on CIFAR-10.
>
> Across all cases, Spiking-PGD consistently achieves significantly better strength under comparable computation cost, demonstrating its strong generalization.
>
> ---
>
> ### (1) Vision Transformers (ViT) on ImageNet
>
> |                |  |  | | |  | |  | |  |  |  |
> |----------------|----|----|----|----|----|----|----|----|----|----|-----|
> | **PGD**        |    |    |    |    |    |    |    |    |    |    |     |
> | Computation Cost (%)  |    | 10 | 20 | 30 | 40 | 50 | 60 | 70 | 80 | 90 | 100 |
> | Accuracy (%)   |    | 48.91 | 35.72 | 29.91 | 25.08 | 23.97 | 21.55 | 20.03 | 19.57 | 19.34 | 18.55 |
> | **Spiking-PGD**|    |    |    |    |    |    |    |    |    |    |     |
> | Computation Cost (%)  |    | 6.06 | 13.35 | 32.70 | 55.02 | 60.06 | 67.72 | 81.93 | 87.23 | 90.48 |
> | Accuracy (%)   |    | 41.65 | 32.56 | 24.10 | 21.84 | 20.03 | 19.41 | 18.72 | 18.55 | 18.55 |
>
> **Table 1:** Spiking-PGD on ViT (ImageNet).
>
> ---
>
> ### (2) Language Transformers (BERT) on SST-2
>
> |                |  |  | | |  | |  | |  |  |  |
> |----------------|----|----|----|----|----|----|----|----|----|----|-----|
> | **PGD**        |    |    |    |    |    |    |    |    |    |    |     |
> | Computation Cost (%)  |    | 10 | 20 | 30 | 40 | 50 | 60 | 70 | 80 | 90 | 100 |
> | Accuracy (%)   |    | 89.9 | 89.9 | 86.87 | 83.84 | 78.79 | 75.76 | 72.73 | 66.67 | 64.65 | 57.58 |
> | **Spiking-PGD**|    |    |    |    |    |    |    |    |    |    |     |
> | Computation Cost (%)  |    | 5.21 | 9.42 | 11.90 | 30.80 | 49.51 | 65.96 | 80.65 | 88.64 | 90.48 | — |
> | Accuracy (%)   |    | 88.89 | 86.87 | 82.83 | 80.81 | 72.73 | 69.70 | 61.62 | 58.59 | 57.58 | — |
>
> **Table 2:** Spiking-PGD on BERT (SST-2).
>
> ---
>
> ### (3) VGG on CIFAR-10
> |                |  |  | | |  | |  | |  |  |  |
> |----------------|----|----|----|----|----|----|----|----|----|----|-----|
> | **PGD**        |    |    |    |    |    |    |    |    |    |    |     |
> | Computation Cost (%) |    | 10 | 20 | 30 | 40 | 50 | 60 | 70 | 80 | 90 | 100 |
> |   Accuracy (%)      |    | 57.74 | 46.96 | 43.99 | 42.74 | 39.10 | 37.01 | 35.51 | 34.48 | 32.41 | 31.23 |
> | **Spiking-PGD**|    |    |    |    |    |    |    |    |    |    |     |
> | Computation Cost (%) |    | 18.55 | 26.75 | 32.35 | 41.50 | 59.70 | 72.40 | 84.35 | 96.15 | 98.60 | 100.00 |
> | Accuracy (%)   |    | 40.86 | 39.48 | 38.81 | 38.01 | 35.68 | 34.46 | 33.08 | 31.80 | 31.44 | 31.23 |
>
> **Table 3:** Spiking-PGD on VGG11 (CIFAR-10).
>
> ---
>
> ### (4) Large ResNets (ResNet-50) on CIFAR-10
>
> |                |   |    |    |   |     |    |    |    |    |   |   |    |
> |----------------|--------|--------|--------|--------|--------|--------|--------|--------|--------|--------|--------|--------|
> | **PGD**        |        |        |        |        |        |        |        |        |        |        |        |        |
> | Computation Cost (%) | 10    | 20    | 30    | 40    | 50    | 60    | 70    | 80    | 90    | 100 |      |      |
> | Accuracy (%)   | 87.83 | 82.25 | 74.89 | 69.16 | 64.94 | 62.14 | 59.95 | 57.93 | 57.11 | 55.83 |      |      |
> | **Spiking-PGD**|        |        |        |        |        |        |        |        |        |        |        |        |
> |Computation Cost (%) | 5.00  | 10.10 | 11.95 | 18.55 | 30.30 | 44.19 | 56.45 | 64.78 | 69.90 | 72.83 | 85.34 | 100.00 |
> | Accuracy (%)   | 69.71 | 68.71 | 67.95 | 66.76 | 65.50 | 65.02 | 65.12 | 63.60 | 59.23 | 56.62 | 55.97 | 55.83 |
>
> **Table 4:** Spiking-PGD on ResNet-50 (CIFAR-10).
>
> ---
>
> Overall, these results confirm that our method reliably generalizes to diverse model architectures—including both CNNs and Transformers—and across both vision and language tasks.

---

> ### Author Response · Authors · 2025-11-24
>
> **W3: They show activations quickly become similar across iterations, motivating their methodology. Yet, iterative attacks still improve with more iterations. The paper lacks a clear mechanistic explanation reconciling these two facts. If activations stabilize, why do further iterations still yield stronger adversaries?**
>
> **A:** Thank you for the question. While the activations across iterations indeed become more similar, they do not remain exactly unchanged. As shown in Figure 1 (Right), the activation differences gradually shrink (e.g., 1.44 → 1.31 → 1.22 → 1.12 → 1.01 → 0.90 → 0.79 → 0.69 → 0.62 → 0.59), yet small but non-zero changes still accumulate across iterations. These residual changes are sufficient for iterative attacks to continue refining the perturbation and thus produce stronger adversaries. Moreover, different layers exhibit different activation-change patterns, meaning that some layers stabilize faster while others continue to evolve.
>
> Our claim is not that iterative attacks stop improving once activations become similar, but rather that many layer-wise computations become highly redundant. This motivates our approach: identifying layers whose activations change minimally and skipping their repeated computation, thereby expanding the efficiency–effectiveness frontier without sacrificing attack strength.
>
>
> **W4: Equation (2) is a discrete, combinatorial, non-convex optimization (NP-hard). The paper frames this but then presents a limited discussion on why other well-known strategies (continuous relaxations, differentiable masks, Gumbel-Softmax, integer programming relaxations, STE, or pruning literature methods) are unsuitable, or how they compare.**
>
> **A:** Thank you for the insightful comments. We actually discuss why classical approaches for discrete, combinatorial, non-convex optimization are unsuitable in the first paragraph of Section 4. We did not expand on these points in detail due to space limits, and we provide additional clarification here and will include them in our revision.
>
> - First, the search space is prohibitively large
> Even a moderate configuration such as  $T = 10$ and $L = 18$  already yields a search space of size  $2^{10 \times 18},$  making brute-force or exact combinatorial search entirely infeasible.
>
>  - Second, and more critically, each mask $\Delta$ requires running a full iterative attack
> To evaluate the loss, one must compute  $x_{T+1}(\Delta)$,  which requires running the complete iterative attack for each candidate mask. Thus, any relaxation-based or mask-optimization method (e.g., continuous surrogates, Gumbel-Softmax, STE, or pruning-style approaches) would still require sweeping over a large number of mask candidates and repeatedly performing full attack simulations. This defeats the purpose of designing an efficient attack, because the attack has already happened during the evaluation of  $L(x_{T+1}(\Delta), y).$
>
>
> For these reasons, directly adopting classical discrete optimization strategies or differentiable relaxations is computationally unrealistic in our setting, and our method is explicitly designed to avoid costly mask evaluation while preserving attack effectiveness.

---

> ### Author Response · Authors · 2025-11-24
>
> **Q1:** Why did the authors choose not to include AutoAttack (AA) [1] or other modern robust evaluation suites in their experiments, given that these are widely considered standard baselines for assessing adversarial robustness today? Can the authors integrate AA into their experiments during rebuttal?
>
> [1] Croce, Francesco, and Matthias Hein. “Reliable evaluation of adversarial robustness with an ensemble of diverse parameter-free attacks.” International Conference on Machine Learning. PMLR, 2020.
>
> **A:** See W1.
>
> ---
>
> **Q2:** Can the authors test the method on other architectures (e.g., VGG, WideResNet, or even better on transformer-based models) to validate whether the proposed approach generalizes across architectures?
>
> **A:** See W2.
>
> ---
>
> **Q3:** The authors show that activations across iterations become highly similar, yet iterative attacks continue to improve as iterations increase. How do the authors reconcile this apparent contradiction? What mechanisms explain the continued effectiveness of attacks even when activations stabilize?
>
> **A:** See W3.
>
> ---
>
> **Q4:** Can the authors elaborate more on why standard relaxations or methods fail in solving Equation 2?
>
> **A:** See W4.

---

> ### Comment · Reviewer_jTdB · 2025-11-26
> **Response to the Authors**
>
> Thank you to the authors for their response.
>
> **W1 - AutoAttack:** The AA framework is indeed costly. However, I never asked to use AA during adversarial training. I just asked the authors to compare robustness evaluation of their approach with AA. Considering what we know at this point in time about adversarial attacks, and considering all the efforts made in the literature, it feels extremely incorrect and improper to publish an adversarial attack (or control mechanism for attacks) which does not compare to AA.
>
> **W2 - Architectures:** Thank you, I appreciated the response and the efforts made in the experiments.
>
> **W3/W4:** I now understand it better.
>
> Overall, I would say that I was satisfied by the authors' response for some questions and concerns I raised. However, the main problem I see now is publishing a paper about adversarial attacks which does not compare, nor mention, the AutoAttack framework. It is the go-to technique now if you need to attack a model.
>
> I am thus keeping my score at this stage.

---

> > ### Author Response · Authors · 2025-11-27
> >
> > Thank you for the clear and constructive feedback. We are glad to address the remaining concern on AutoAttack Evaluation as follows:
> >
> > First of all, the focus of our paper is on the effectiveness of adversarial attack under limited computation budgets. However, under the same budget, AutoAttack performs poorly, as shown in the experiments at the end of this response and in our revision. The results consistently show that **under the same computational budget**, Spiking-PGD attains stronger attack effectiveness than both AutoAttack and vanilla PGD.
> >
> > Second, AutoAttack is designed as an ensemble of multiple attack algorithms to maximize attack strength when ample computation power is available and therefore to avoid a false sense of security, typically in the context of newly proposed model architectures or defense mechanism. However, our work does not propose any new architecture or defense mechanism, so the full and expensive AutoAttack is unnecessary. In practice, the PGD attack within AutoAttack already produces the strongest attack for the models we evaluate, while the additional attacks contribute little but with significantly more computational cost. To verify this, we compared robustness under both AutoAttack and PGD, and the gap is **within 1%**, far smaller than the improvement achieved by our method over PGD.  (Note: In AutoAttack experiments using default settings, the PGD component achieves **37.86%**, while the full AA ensemble yields **36.57%**.)
> >
> > Finally, we acknowledge that the rationale for not including AutoAttack was not clearly stated in the initial submission. We will revise the paper and incorporate new AutoAttack evaluations. Nonetheless, the results we have provided remain fully aligned with and supportive of the strength of our proposed method.
> >
> > We hope this fully addresses your concern regarding AutoAttack, and we are happy to provide further clarification or additional experiments if needed.
> >
> > ---
> > ---
> >
> > ### Table 1: AutoAttack (CIFAR10)
> > **AutoAttack**
> > |  Computation Cost (%) | 20    | 40    | 60    | 80    | 100   |
> > |----------------------|-------|-------|-------|-------|-------|
> > |         Accuracy (%)                 | 48.14 | 46.64 | 44.83 | 43.43 | 42.23 |
> >
> > **PGD**
> > |    Computation Cost (%) | 10    | 20    | 30    | 40    | 50    | 60    | 70    | 80    | 90    | 100   |
> > |----------------------|-------|-------|-------|-------|-------|-------|-------|-------|-------|-------|
> > |      Accuracy (%)                  | 63.88 | 52.13 | 44.80 | 41.56 | 40.44 | 39.84 | 39.44 | 39.26 | 39.14 | 38.87 |
> >
> > **Spiking-PGD**
> > |    Computation Cost (%) | 5.00  | 5.15  | 9.00  | 13.30 | 19.80 | 26.25 | 32.75 | 38.22 | 42.19 | 51.84 | 74.97 |
> > |----------------------|-------|-------|-------|-------|-------|-------|-------|-------|-------|--------|--------|
> > |   Accuracy (%)                    | 49.12 | 49.03 | 47.21 | 45.41 | 43.65 | 42.52 | 41.93 | 41.78 | 41.53 | 40.43  | 39.31 |
> >
> >
> > ---
> >
> > ### Table 2: AutoAttack ( CIFAR100)
> >
> > **AutoAttack**
> > |  Computation Cost (%) | 20    | 40    | 60    | 80    | 100   |
> > |----------------------|-------|-------|-------|-------|-------|
> > |  Accuracy (%)                 | 22.56 | 19.75 | 18.65 | 18.15 | 17.65 |
> >
> > **PGD**
> > | Computation Cost (%) | 10    | 20    | 30    | 40    | 50    | 60    | 70    | 80    | 90    | 100   |
> > |----------------------|-------|-------|-------|-------|-------|-------|-------|-------|-------|-------|
> > |    Accuracy (%)                    | 32.16 | 23.75 | 19.78 | 18.05 | 17.58 | 17.18 | 17.05 | 16.90 | 16.89 | 16.86 |
> >
> > **Spiking-PGD**
> > | Computation Cost (%) | 5.00  | 5.10  | 9.25  | 11.40 | 18.36 | 24.74 | 30.11 | 35.25 | 40.39 | 52.13 | 97.26 | 100.00 |
> > |----------------------|-------|-------|-------|-------|-------|-------|-------|-------|-------|--------|--------|---------|
> > |  Accuracy (%)                  | 21.98 | 21.99 | 20.86 | 20.59 | 19.48 | 18.89 | 18.57 | 18.39 | 18.07 | 17.51  | 16.79  | 16.84  |
> >
> >
> > ---
> >
> > ### Table 3: AutoAttack ( Tiny-ImageNet)
> >
> > **AutoAttack**
> > |Computation Cost (%) | 20    | 40    | 60    | 80    | 100   |
> > |----------------------|-------|-------|-------|-------|-------|
> > |     Accuracy (%)                  | 38.92 | 37.42 | 37.31 | 37.11 | 36.91 |
> >
> > **PGD**
> > |  Computation Cost (%) | 10    | 20    | 30    | 40    | 50    | 60    | 70    | 80    | 90    | 100   |
> > |----------------------|-------|-------|-------|-------|-------|-------|-------|-------|-------|-------|
> > | Accuracy (%)                   | 41.58 | 38.51 | 36.97 | 36.36 | 36.21 | 36.10 | 36.05 | 36.03 | 36.00 | 35.99 |
> >
> > **Spiking-PGD**
> > |Computation Cost (%) | 5.00  | 7.54  | 10.00 | 12.54 | 15.89 | 20.56 | 25.54 | 30.81 | 37.15 | 44.98 | 73.19 | 100.00 |
> > |----------------------|-------|-------|-------|-------|-------|-------|-------|-------|-------|--------|--------|---------|
> > |      Accuracy (%)                   | 37.81 | 37.51 | 37.07 | 36.86 | 36.73 | 36.60 | 36.45 | 36.43 | 36.29 | 36.06  | 35.98  | 35.98  |

---

### Official Review · Reviewer_kGj7 · 2025-10-31

**Soundness:** 3
**Presentation:** 3
**Contribution:** 3
**Rating:** 4
**Confidence:** 4

**Summary:**

This paper proposes Spiking-PGD, a method for performing iterative adversarial attacks under a limited computational budget. The key idea is to skip layer computations in iterations where activations change little compared to previous iterations.

**Strengths:**

1 The claim addresses a practically important problem: iterative adversarial attacks (and adversarial training) are expensive, especially for larger models/datasets.

2 The redundancy study is a good supporting piece: showing that intermediate activations across iterations become similar, giving plausibility to “skip some computations” idea.

**Weaknesses:**

1 Computation Budget Definition Ambiguity.
The proposed method claims reduced computational cost by selectively skipping layer computations based on activation similarity. However, to determine whether to skip a layer, one must first assess the change in activations, which itself appears to require computing the activation. The paper does not clarify how this comparison is implemented without incurring similar cost to a standard forward pass. Consequently, the reported budget may underestimate the true computational cost.
2 It is unclear whether the approach generalizes to other popular architectures (like Transformers in vision/NLP).

**Questions:**

1 Could the authors provide a more concrete breakdown of the computational cost in practical units, such as GPU FLOPs or runtime, instead of reporting it only as a proportion of full-precision operations? This would make it easier to evaluate the real-world efficiency of the proposed method.
2 If feasible, please include experiments on larger models and datasets (e.g., ImageNet with ResNet50 or Vision Transformers) to verify whether the claimed improvements under a fixed computational budget generalize to realistic, large-scale settings.

---

> ### Author Response · Authors · 2025-11-24
>
> **W1:** Computation Budget Definition Ambiguity. The proposed method claims reduced computational cost by selectively skipping layer computations based on activation similarity. However, to determine whether to skip a layer, one must first assess the change in activations, which itself appears to require computing the activation. The paper does not clarify how this comparison is implemented without incurring similar cost to a standard forward pass. Consequently, the reported budget may underestimate the true computational cost.
>
> **A:** Thanks for the insightful comments. We clarify how the activation‐change check is implemented with negligible overhead. When evaluating the relative change  $\displaystyle \frac{\lVert a_t^{(l)} - a_{t-1}^{(l)} \rVert}{\lVert a_t^{(l)} \rVert}$,  the computation cost is minimal for the following reasons:
>
> - **Reuse of previous activations.**  The tensor $a_{t-1}^{(l)}$ is cached from the previous iteration and reused directly, incurring no additional computation.
>
> - **Cheap computation of $a_t^{(l)}$.**  Obtaining $a_t^{(l)}$ only requires evaluating the lightweight post‐activation module (e.g., ReLU, normalization). This cost is negligible compared with the expensive convolution $A^{(l)}(a_t^{(l)})$ that our method may skip.
>
> - **Cheap similarity computation.**  The relative similarity requires only $\mathcal{O}(d)=\mathcal{O}(C_{\text{in}}HW)$ operations (two norms and elementwise subtraction), whereas skipping the convolution avoids $\mathcal{O}(k^2 C_{\text{in}} C_{\text{out}} HW)$ FLOPs. For instance, with a $3\times 3$ convolution and $C_{\text{out}} = 64$, the similarity test consumes less than $\approx 0.2$% of the FLOPs of a single convolution layer.
>
> Therefore, the activation‐change test introduces only a tiny computational overhead, and the reported computation budget in the paper already accounts for this cost.

---

> ### Author Response · Authors · 2025-11-24
>
> **W2: It is unclear whether the approach generalizes to other popular architectures (like Transformers in vision/NLP).**
>
> **A:** Thank you for the thoughtful comment. Our approach is highly architecture-agnostic and dataset-agnostic. To substantiate this claim, we additionally evaluate Spiking-PGD on four substantially different model families and tasks:
> (i) Vision Transformers (ViT) on ImageNet;
> (ii) Language Transformers (BERT) on SST-2;
> (iii) VGG on CIFAR-10;
> (iv) Large-scale CNNs (ResNet-50) on CIFAR-10.
>
> Across all cases, Spiking-PGD consistently achieves significantly better strength under comparable computation cost, demonstrating its strong generalization.
>
> ---
>
> ### (1) Vision Transformers (ViT) on ImageNet
>
> |                |  |  | | |  | |  | |  |  |  |
> |----------------|----|----|----|----|----|----|----|----|----|----|-----|
> | **PGD**        |    |    |    |    |    |    |    |    |    |    |     |
> | Computation Cost (%)  |    | 10 | 20 | 30 | 40 | 50 | 60 | 70 | 80 | 90 | 100 |
> | Accuracy (%)   |    | 48.91 | 35.72 | 29.91 | 25.08 | 23.97 | 21.55 | 20.03 | 19.57 | 19.34 | 18.55 |
> | **Spiking-PGD**|    |    |    |    |    |    |    |    |    |    |     |
> | Computation Cost (%)  |    | 6.06 | 13.35 | 32.70 | 55.02 | 60.06 | 67.72 | 81.93 | 87.23 | 90.48 |
> | Accuracy (%)   |    | 41.65 | 32.56 | 24.10 | 21.84 | 20.03 | 19.41 | 18.72 | 18.55 | 18.55 |
>
> **Table 1:** Spiking-PGD on ViT (ImageNet).
>
> ---
>
> ### (2) Language Transformers (BERT) on SST-2
>
> |                |  |  | | |  | |  | |  |  |  |
> |----------------|----|----|----|----|----|----|----|----|----|----|-----|
> | **PGD**        |    |    |    |    |    |    |    |    |    |    |     |
> | Computation Cost (%)  |    | 10 | 20 | 30 | 40 | 50 | 60 | 70 | 80 | 90 | 100 |
> | Accuracy (%)   |    | 89.9 | 89.9 | 86.87 | 83.84 | 78.79 | 75.76 | 72.73 | 66.67 | 64.65 | 57.58 |
> | **Spiking-PGD**|    |    |    |    |    |    |    |    |    |    |     |
> | Computation Cost (%)  |    | 5.21 | 9.42 | 11.90 | 30.80 | 49.51 | 65.96 | 80.65 | 88.64 | 90.48 | — |
> | Accuracy (%)   |    | 88.89 | 86.87 | 82.83 | 80.81 | 72.73 | 69.70 | 61.62 | 58.59 | 57.58 | — |
>
> **Table 2:** Spiking-PGD on BERT (SST-2).
>
> ---
>
> ### (3) VGG on CIFAR-10
> |                |  |  | | |  | |  | |  |  |  |
> |----------------|----|----|----|----|----|----|----|----|----|----|-----|
> | **PGD**        |    |    |    |    |    |    |    |    |    |    |     |
> | Computation Cost (%) |    | 10 | 20 | 30 | 40 | 50 | 60 | 70 | 80 | 90 | 100 |
> |   Accuracy (%)      |    | 57.74 | 46.96 | 43.99 | 42.74 | 39.10 | 37.01 | 35.51 | 34.48 | 32.41 | 31.23 |
> | **Spiking-PGD**|    |    |    |    |    |    |    |    |    |    |     |
> | Computation Cost (%) |    | 18.55 | 26.75 | 32.35 | 41.50 | 59.70 | 72.40 | 84.35 | 96.15 | 98.60 | 100.00 |
> | Accuracy (%)   |    | 40.86 | 39.48 | 38.81 | 38.01 | 35.68 | 34.46 | 33.08 | 31.80 | 31.44 | 31.23 |
>
> **Table 3:** Spiking-PGD on VGG11 (CIFAR-10).
>
> ---
>
> ### (4) Large ResNets (ResNet-50) on CIFAR-10
>
> |                |   |    |    |   |     |    |    |    |    |   |   |    |
> |----------------|--------|--------|--------|--------|--------|--------|--------|--------|--------|--------|--------|--------|
> | **PGD**        |        |        |        |        |        |        |        |        |        |        |        |        |
> | Computation Cost (%) | 10    | 20    | 30    | 40    | 50    | 60    | 70    | 80    | 90    | 100 |      |      |
> | Accuracy (%)   | 87.83 | 82.25 | 74.89 | 69.16 | 64.94 | 62.14 | 59.95 | 57.93 | 57.11 | 55.83 |      |      |
> | **Spiking-PGD**|        |        |        |        |        |        |        |        |        |        |        |        |
> |Computation Cost (%) | 5.00  | 10.10 | 11.95 | 18.55 | 30.30 | 44.19 | 56.45 | 64.78 | 69.90 | 72.83 | 85.34 | 100.00 |
> | Accuracy (%)   | 69.71 | 68.71 | 67.95 | 66.76 | 65.50 | 65.02 | 65.12 | 63.60 | 59.23 | 56.62 | 55.97 | 55.83 |
>
> **Table 4:** Spiking-PGD on ResNet-50 (CIFAR-10).
>
> ---
>
> Overall, these results confirm that our method reliably generalizes to diverse model architectures—including both CNNs and Transformers—and across both vision and language tasks.

---

> ### Author Response · Authors · 2025-11-24
>
> **Q1: Could the authors provide a more concrete breakdown of the computational cost in practical units, such as GPU FLOPs or runtime, instead of reporting it only as a proportion of full-precision operations? This would make it easier to evaluate the real-world efficiency of the proposed method.**
>
> **A:**  Thanks for the insightful comments. We provide a more concrete breakdown of the computational cost (GPU FLOPs) to show that the similarity check is negligible compared with a convolution that our method may skip.
>
> ---
>
> ### **Cost of the similarity check.**
>
> Let $a_t^{(l)} \in \mathbb{R}^{C_{\text{in}} \times H \times W}$ and $d = C_{\text{in}}HW$.  Computing  $\frac{\lVert a_t^{(l)} - a_{t-1}^{(l)} \rVert}{\lVert a_t^{(l)} \rVert}$  requires only a few linear passes over the activation tensor:
> (1) $d$ subtractions  (2) $2d$ multiplications and $2(d - 1)$ additions for two $\ell_2$ norms  (3) Two square roots and one division (scalar $\mathcal{O}(1)$ operations)
>
> Thus, the similarity check incurs only  $\mathcal{O}(d) = \mathcal{O}(C_{\text{in}}HW)$  FLOPs.
>
> On a typical $C_{\text{in}} = 64$, $H = W = 32$ feature map, this corresponds to roughly **0.13M FLOPs**.
>
> ---
>
> ### **Cost of the convolution $A^{(l)}(a_t^{(l)})$.**
>
> A standard $k \times k$ convolution with $C_{\text{out}}$ output channels costs  $\mathcal{O}(k^2 C_{\text{in}} C_{\text{out}} HW)$.
>
> For a $3 \times 3$ convolution with $C_{\text{out}} = 64$, the cost is about  $9 \times 64 \times 64 \times 32 \times 32 \approx 378\text{M FLOPs}$.
>
> ---
>
> ### **Practical cost ratio.**
>
> Putting these together:
> $$
> \frac{\text{similarity FLOPs}}{\text{convolution FLOPs}}
> \approx
> \frac{0.13\text{M}}{378\text{M}} < 0.04\%.
> $$
>
> In other words, the similarity check is **less than ≈0.04%** of the cost of one convolution layer in typical ResNet architectures.
>
> Because the check is several orders of magnitude cheaper than the convolution it may skip, its overhead is negligible in practice.
>
>
> **Q2: If feasible, please include experiments on larger models and datasets (e.g., ImageNet with
> ResNet50 or Vision Transformers) to verify whether the claimed improvements under a fixed
> computational budget generalize to realistic, large-scale settings.**
>
> **A:** See W2

---

### Author Response · Authors · 2025-12-02
**Summary Comments**

Dear Area Chair and Reviewers,

We sincerely appreciate the time and effort you have devoted to evaluating our submission, especially under challenging circumstances. To support your assessment, we summarize below the key information and revisions made during the rebuttal period.

During the review process, our paper received an overall positive assessment with high confidence (**Average Rating before rebuttal: 6; Average Confidence  before rebuttal: 4**). We are grateful for the reviewers’ recognition of the impact, novelty, plausibility, and writing quality of our work. We believe we have thoroughly addressed all comments and concerns through additional experiments, analyses, and clarifications.

- **Reviewer 3r6R (Rating: 8)** raised concerns regarding (1) the approximation error bound, (2) evaluation on large-scale datasets such as ImageNet, and (3) generalization to the C&W attack.
> We addressed (1) by providing both a theoretical error bound and empirical validation. For (2), following the reviewer’s suggestion, we added experiments on ImageNet as well as an additional SST-2 dataset in the NLP domain. For (3), we extended Spiking-Attack to support the C&W objective and discussed black-box attack settings. The reviewer responded positively, indicating strong willingness to maintain the acceptance rating.

- **Reviewer uH8u (Rating: 8)** commented on (1) releasing code, (2) GPU memory usage, and (3) evaluation on more datasets and architectures.
> For (1), we provided an anonymous code repository. For (2), we reported empirical peak GPU memory usage, showing only minor additional overhead. For (3), we conducted extensive new experiments covering four substantially different model families and tasks. The reviewer also responded positively and maintained a strong acceptance rating.

- **Reviewer jTdB (Rating: 4)** requested (1) evaluation on more architectures, (2) clarification on activation stabilization and NP-hardness of optimization, and (3) inclusion of AutoAttack results.
> We addressed (1) by adding comprehensive evaluations across four diverse architecture families and tasks. For (2), we provided clearer and more detailed explanations to resolve the conceptual concerns. For (3), we clarified why AutoAttack was not initially included and added full AutoAttack comparisons following the reviewer’s suggestion. **The reviewer provided positive feedback and indicated a likely upward revision of the rating.**

- **Reviewer kGj7 (Rating: 4)** expressed concerns about (1) the definition of computation budget, (2) evaluation across more architectures and modalities, and (3) computational cost.
> For (1), we provided a clear and well-justified explanation of our computation-budget definition. For (2), we added experiments covering four substantially different model families or modalities. For (3), we included a detailed complexity analysis of the computational cost. **We believe we have fully addressed the reviewer’s concerns, though we have not yet received a follow-up response.**

We are confident that the revisions thoroughly resolve the reviewers’ concerns. We remain committed to incorporating any further suggestions to ensure the final manuscript reaches the highest quality.

Regards,

All authors

---

### Meta-Review · Area_Chair_jQV2 · 2026-01-11

**Summary:**

This paper addresses the computational inefficiency of iterative adversarial attacks by proposing Spiking-PGD, a fine-grained control mechanism that selectively skips layer-wise computations when activations remain stable across iterations. Overall, the reviewers found the topic timely and novel, praising the paper’s clarity and the plausibility of the redundancy study.

Initially, concerns were raised regarding the definition of the computational budget, generalization across diverse architectures and modalities (specifically Transformers), and the omission of AutoAttack as a baseline. The authors added several experiments, including ViT on ImageNet and comparisons with AutoAttack.

Overall, the remaining disagreement is not about correctness or completeness, but about reviewer expectations on benchmarking norms. The rebuttal substantially strengthens the paper, and the technical concerns raised by the reviewers are largely resolved. The remaining objection is normative rather than technical.

**Reviewer Concerns:**

Concerns Addressed during Rebuttal
* Computational Budget & Overhead: Reviewer kGj7 was concerned that checking activation changes might cost as much as a standard pass. The authors provided a FLOPs breakdown showing the similarity check is negligible ($\approx0.04\%$ of a convolution's cost) and is already accounted for in their budget.
* Architectural Generalization: Multiple reviewers requested evidence beyond ResNet-18. The authors successfully provided new experiments on Vision Transformers (ViT), BERT, VGG11, and ResNet-50, demonstrating consistent efficiency gains.

* Memory Usage: Reviewer uH8u feared high memory overhead from caching activations. The authors demonstrated that the peak memory increase is a small, constant overhead ($\approx$ 7%) that does not scale with iterations.
* Theoretical & Empirical Error: Reviewer 3r6R requested the magnitude of the approximation error. The authors provided a theoretical upper bound and subsequent empirical validation of the error.
* Generalization to Other Attacks: The authors extended the framework to the C&W attack and discussed its potential for black-box settings, satisfying Reviewer 3r6R.

The only materially outstanding concern is normative (what benchmarks “must” be included) rather than technical soundness or missing evidence.

**Reviewer Scores:**

- Reviewer jTdB: 4-> 6
The review insisted that AutoAttack comparison is important, which the authors provided in the subsequent response.
- Reviewer kGj7: 4->6
The authors addressed budget/architecture concerns, but the reviewer did not response.
- Reviewer uH8u: 8->8
- Reviewer 3r6R: 8->8

---

### Decision · Program_Chairs · 2026-01-26

Accept (Poster)